# Directed natural evolution generates a next-generation oncolytic virus with a high potency and safety profile

Li Guo[1,7], Cheng Hu[2,7], Yang Liu[1], Xiaoyu Chen[1], Deli Song[1], Runling Shen[1], Zhanzhen Liu[3], Xudong Jia [4], Qinfen Zhang[4], Yuanzhu Gao[4], Zhezhi Deng[5], Tao Zuo [3], Jun Hu[1], Wenbo Zhu[1], Jing Cai[1], Guangmei Yan[1], Jiankai Liang [1] ✉ & Yuan Lin [1,6] ✉

Oncolytic viruses (OVs) represent a type of encouraging multi-mechanistic drug for the treatment of cancer. However, attenuation of virulence, which is generally required for the development of OVs based on pathogenic viral backbones, is frequently accompanied by a compromised killing effect on tumor cells. By exploiting the property of viruses to evolve and adapt in cancer cells, we perform directed natural evolution on refractory colorectal cancer cell HCT-116 and generate a next-generation oncolytic virus M1 (NGOVM) with an increase in the oncolytic effect of up to 9690-fold. The NGOVM has a broader antitumor spectrum and a more robust oncolytic effect in a range of solid tumors. Mechanistically, two critical mutations are identified in the E2 and nsP3 genes, which accelerate the entry of M1 virus by increasing its binding to the Mxra8 receptor and antagonize antiviral responses by inhibiting the activation of PKR and STAT1 in tumor cells, respectively. Importantly, the NGOVM is well tolerated in both rodents and nonhuman primates. This study implies that directed natural evolution is a generalizable approach for developing next-generation OVs with an expanded scope of application and high safety.

Oncolytic virotherapy is an emerging treatment modality for advanced malignancies that uses replicating viruses. These natural or genetically engineered viruses exhibit tumor-selective replication and killing, desirable immunogenic properties, and targeted delivery of therapeutic genes to tumors. IMLYGIC, the first OV approved by the FDA in 2015, is a genetically modified oncolytic herpes simplex virus type 1

(HSV-1) used to treat advanced melanoma[1,2]. Currently, most OVs are based on human pathogenic viruses, such as HSV and adenoviruses; thus, attenuation of virulence by removal of toxic viral genes is generally required. However, the decrease in viral pathogenicity is frequently, if not always, accompanied by impairment of viral replication in tumor cells, which is indispensable for direct oncolysis and immune

[1]Department of Pharmacology, Department of Microbiology, Zhongshan School of Medicine, Sun Yat-sen University, Guangzhou 510080, China. [2]Department of Urology, The Third Affiliated Hospital of Sun Yat-sen University, Guangzhou 510630, China. [3]Department of Colorectal Surgery, The Sixth Affiliated Hospital, Sun Yat-sen University, Guangzhou, Guangdong, P. R. China and Guangdong Provincial Key Laboratory of Colorectal and Pelvic Floor Diseases, The Sixth Affiliated Hospital, Sun Yat-sen University, Guangzhou 510630, China. [4]School of Life Sciences, Sun Yat-sen University, Guangzhou 510275, China. [5]Department of Neurology, The First Affiliated Hospital, Sun Yat-sen University, Guangdong Provincial Key Laboratory of Diagnosis and Treatment of Major Neurological Diseases, National Key Clinical Department and Key Discipline of Neurology, Guangzhou 510080, China. [6]Advanced Medical Technology Center, The First Affiliated Hospital-Zhongshan School of Medicine, Sun Yat-sen University, Guangzhou, China. [7]These authors contributed equally: Li Guo, Cheng Hu. ✉e-mail: liangjk5@mail.sysu.edu.cn; liny96@mail.sysu.edu.cn

activation[3]. In 2021, a third-generation oncolytic HSV-1, DELYTACT, with greater replication ability in tumor cells, received conditional and time-limited marketing approval in Japan for the treatment of malignant glioma based on the results of a phase 2 clinical trial with 19 enrolled patients[4,5]. Therefore, discovering or engineering novel oncolytic viral backbones with improved selective viral replication in tumor cells and high safety in normal cells is urgently needed.

In general, tumor-selective viral replication is fundamental to direct oncolysis and the subsequent activation of the antitumor immune response[6]. Nevertheless, in refractory tumor cells, the replication of OVs is severely inhibited. Hence, it is essential to improve the infectivity of OVs to increase the response rate. Although their replication is limited in refractory tumor cells, OVs can still replicate moderately in these cells, providing the possibility of exploiting evolution by serial passage, which has widespread application in virology research, to generate selective and powerful OVs[3]. By using this approach, with the accumulation of adaptive mutations, OVs with potent activity against refractory tumor cells may be generated. There are a few reports showing that the replication of vesicular stomatitis virus (VSV) was improved by serial passage in Her2/neu-expressing breast cancer cells, glioblastoma cells, and pancreatic ductal adenocarcinoma (PDAC) cells. However, the infectivity of these evolved VSVs is maintained or even further attenuated in nonmalignant cells[7–9]. These studies suggest that the application of directed evolution strategies could make OVs adapt to various heterogeneous tumor cells without damaging normal cells.

M1 is an enveloped positive-strand RNA alphavirus with a genome encoding four nonstructural proteins (nsP1-nsP4) and five structural proteins (capsid-E3-E2-6k-E1). Our group identified and characterized the natural alphavirus M1 as an effective and safe OV[10,11]. Based on the understanding of M1 virus, we successfully reinforced the anticancer activity of OV M1 by combining a series of small molecule compounds or immune checkpoint antibodies[12–19]. However, one of the drawbacks of combination therapy is that the targets or regulated pathways of the other drug may be complex, which may potentially affect the replication or safety of the OV. Therefore, the use of a directed evolution strategy based on natural selection to endow OVs themselves with more potent tumoricidal efficacy may be a better choice for optimizing and generating next-generation OVs.

In this work, we adapt OV M1 to refractory colorectal carcinoma cell line HCT-116 and obtain NGOVM, a potent oncolytic virus for solid cancers. We find two critical mutations in NGOVM, one in E2 protein that enhances receptor binding and viral entry, and another in nsP3 protein that inhibits PKR-STAT1 antiviral pathway and boosts viral replication. NGOVM shows increased oncolytic efficacy and a high safety profile in various models.

## Results
### Directed natural evolution identifies a potentiated strain of M1 virus
To enhance the oncolytic efficacy of M1 virus by directed evolution, we performed serial passage in the refractory colorectal carcinoma cell line HCT-116. For real-time observation and quantification of virus replication, we used a recombinant M1 virus expressing green fluorescent protein (GFP) during virus replication, which was called M1-GFP[20]. HCT-116 cells were infected with M1-GFP (termed P0) at an MOI of 10. Seventy-two hours after infection, the culture supernatant was collected for the next passage (Fig. 1a). At passage 3 (P3), the proportion of GFP-positive cells was obviously increased, indicating that the infection, replication or spread of M1 virus in HCT-116 cells was enhanced, possibly owing to the generation of HCT-116 cell-adapted variants of M1 virus. With continued passaging, the enhanced infectivity and killing capacity of these M1 variants were maintained (Fig. 1b). We next compared the oncolytic efficacy of the adapted variants with that of the parental virus by an MTT assay. The dose-

response curves showed that the oncolytic efficacy of the M1 variants was substantially potentiated (Fig. 1c). The half-maximal effective concentration (EC50) was determined using nonlinear regression. Compared with that of M1-GFP, the EC50 values of the M1 variants were reduced by 6470- to 6970-fold (Fig. 1c).

Genomic analysis of the variants detected at P3, P8 and P10 identified two missense mutations in M1 variants, namely, A1072C in the nucleotide sequence of nonstructural protein 3 (nsP3) and A12C in the nucleotide sequence of envelope protein E2, which resulted in a methionine (M) to leucine (L) substitution at position 358 in nsP3 and a lysine (K) to asparagine (N) substitution at position 4 in E2 (Fig. 1d). Both mutations were introduced into the M1-GFP backbone by site-directed mutagenesis to generate M1-N3E2M (Fig. 1a), which replicated abundantly in HCT-116 cells and induced a marked cytopathic effect (CPE) (Fig. 1e). The maximum infection rate of M1-N3E2M in HCT-116 cells was 92%, much higher than that of M1-GFP (Figs. 1f, S1). Consistent with this finding, the viral yield of M1-N3E2M was more than $10^3$-fold higher (Fig. 1g). The increased viral replication resulted in enhancement of the oncolytic efficacy, with a reduction in the EC50 of 6690-fold, which was similar to that of M1 variants obtained in the serial passage assay (Fig. 1h). In summary, we developed M1-N3E2M, with encouraging replication ability and oncolytic efficacy, by directed natural evolution.

### M1-N3E2M has a broader antitumor spectrum in vitro
We next sought to determine whether the oncolytic effect of NGOVM is specifically enhanced in HCT-116 cells or broadly enhanced in other types of tumor cells. The oncolytic efficacy of M1-N3E2M was tested in 57 human tumor cell lines, including colorectal carcinoma, liver cancer, pancreatic cancer, breast cancer, prostate cancer, and bladder cancer cell lines. M1-N3E2M exhibited enhanced oncolysis at an MOI of 10 in most of the tested cell lines (Fig. 2a), and most of these were digestive system and breast cancer cell lines, with only a few bladder cancer or prostate cancer cell lines (Fig. 2b). To confirm the enhancement of oncolysis in digestive system tumors, we further evaluated the oncolytic effect of M1-N3E2M in HCT-8, another colorectal carcinoma cell line, and Huh 7, a hepatocellular carcinoma cell line. Extensive replication of M1-N3E2M and an apparent CPE were observed in HCT-8 cells (Fig. 2c), along with a 2760-fold decrease in the EC50, indicating the augmented killing ability (Fig. 2d). Similar results were obtained in Huh 7 cells, the EC50 reduction was 158-fold (Fig. 2e, f).

More importantly, despite the substantial potentiation of replication and killing effects in tumor cells, neither M1-GFP nor M1-N3E2M replicated well in normal human colon fibroblast cell line CCD-18Co (Fig. 2g). Consequently, neither M1-GFP nor M1-N3E2M displayed cytotoxicity in the normal cell line CCD-18Co (Fig. 2h). We further found that the normal cell line tested did not exhibit significant cytotoxicity after exposure to either virus at 100 MOI, which is 10-fold higher than the highest MOI used in cancer cells (Fig. S2). These findings indicate the potent tumor-selective oncolytic effect of M1-N3E2M in a broad range of human cancer cells.

### M1-N3E2M has stronger ability to inhibit tumor growth in vivo and ex vivo
The potent oncolytic effect of M1-N3E2M in vitro prompted us to further investigate its therapeutic potential in vivo. Nude mice were inoculated subcutaneously with HCT-116 cells and injected intravenously with vehicle control, M1-GFP or M1-N3E2M after randomization (Fig. 3a). Consistent with the in vitro results, M1-N3E2M significantly delayed the growth of subcutaneous xenograft tumors, whereas M1-GFP barely suppressed tumor growth (Fig. 3b). During the entire observation period, the mice did not show obvious abnormalities, and their body weights were not significantly different (Fig. S3a). At 24 h after the last treatment, the replication of M1 viruses and the

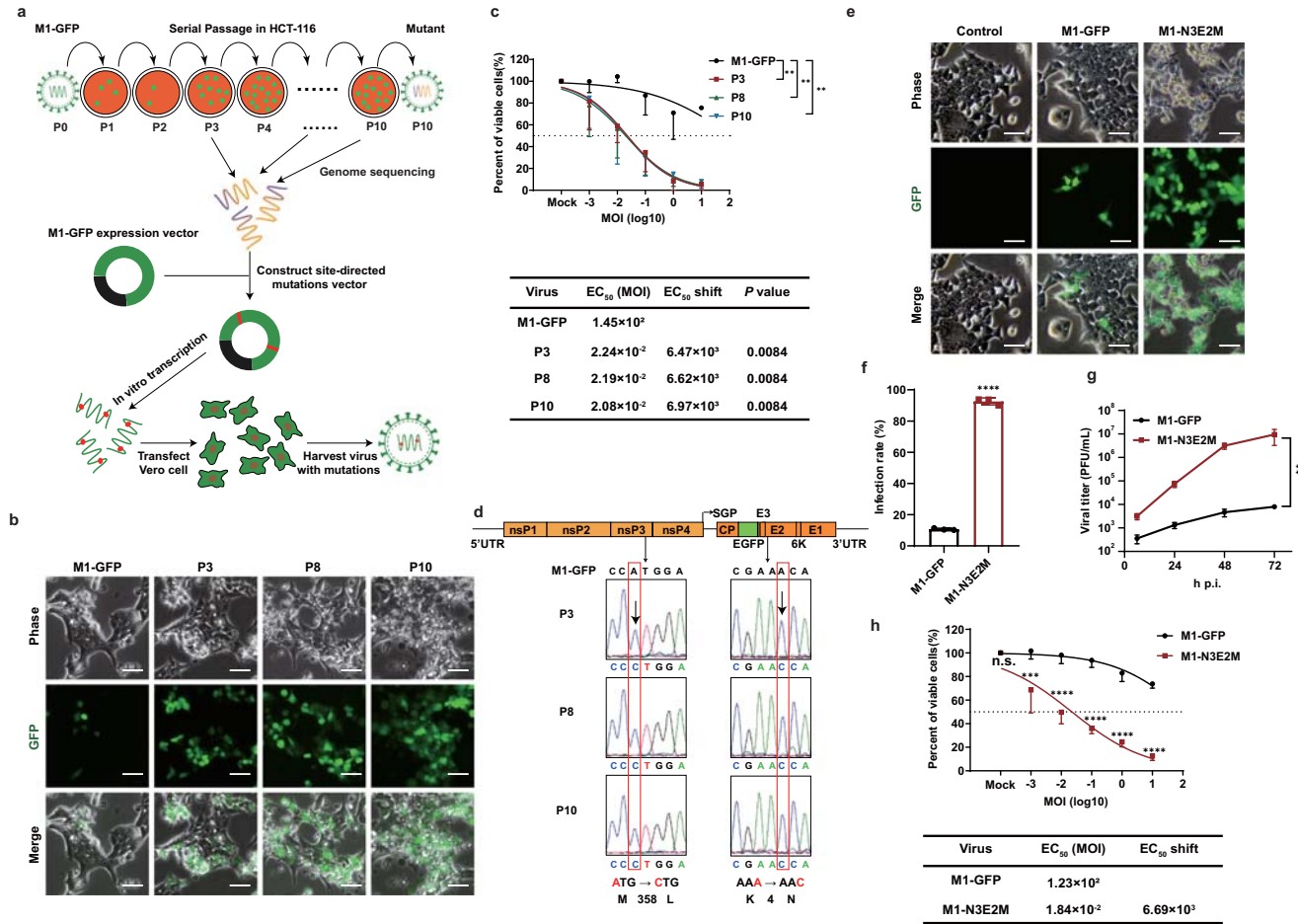

**Fig. 1 | M1 virus was serially passaged in the refractory HCT-116 cell line to generate more effective OVs. a** Schematic of serial passaging in the HCT-116 cell line and generation of mutated viruses by site-directed mutagenesis. **b** Phase contrast and GFP fluorescence images of P3, P8 and P10 are shown. Scale bars, 50 μm. Representative images of $n$ = 3. **c** Cell viability was evaluated by an MTT assay after cells were infected with serial dilutions of M1-GFP, P3, P8 and P10. EC50 shift was calculated by nonlinear regression and the EC50 values were used for statistical analysis by one-way ANOVA with Tukey's multiple comparisons test relative to M1-GFP and adjusted $P$ values are indicated. **d** Schematic of M1-GFP genomic RNA. Viral genomic RNA was isolated for genetic analysis. Nucleotide substitutions are highlighted in red boxes. **e** HCT-116 cells were infected at an MOI of 0.1 and imaged with a phase contrast microscope and a fluorescence microscope 72 h after infection.

Representative images of $n$ = 3. Scale bars, 50 μm. **f** HCT-116 cells were infected with M1-GFP and M1-N3E2M (MOI = 1) for 48 hours, and the infection rate was determined by flow cytometry. $P < 0.0001$ was calculated with Two-tailed unpaired t-test. **g** After infection of HCT-116 cells with M1-GFP and M1-N3E2M at an MOI of 0.1, the viral titer was tested by the CCID50 method. $P = 0.0060$ was determined by Two-way ANOVA relative to M1-GFP. **h** Cell viability was evaluated after cells were infected with serial dilutions of M1-GFP and M1-N3E2M. EC50 shift was calculated by nonlinear regression. Statistical significance was calculated using Two-way ANOVA with Bonferroni's multiple comparisons test relative to M1-GFP. Adjusted $P$ values are: MOI (Mock), $P > 0.9999$; MOI (−3), $P = 0.0003$; MOI (−2 to 1), $P < 0.0001$. n.s.: no significance, $**P < 0.01$, $***P < 0.001$, $****P < 0.0001$. Data are shown as mean ± SD, for $n$ = 3 biological replicates. Source data are provided as a Source Data file.

proliferation and apoptosis of tumor cells were examined by immunohistochemical (IHC) staining. The higher IHC staining intensity of GFP and cleaved caspase3 and lower Ki-67 signal in the M1-N3E2M group than in the M1-GFP group indicated that M1-N3E2M replicated more robustly in tumor tissues, thus inhibiting the proliferation of tumor cells and promoting their apoptosis (Fig. 3c). In normal mouse tissues, viral replication was not detectable, and no visible changes in tissue morphology were observed (Fig. 3d).

To explore whether prolonged and higher-dose administration could more effectively inhibit the growth of tumors in vivo, we established a subcutaneous colorectal tumor model with the HCT-116 cell line and administered vehicle control, M1-GFP or M1-N3E2M intravenously for 21 consecutive days (Fig. 3e). M1-GFP had a moderate inhibitory effect on tumor growth, but M1-N3E2M was markedly more effective at reducing the tumor burden (Fig. 3f). Similarly, the mice remained asymptomatic and showed no obvious difference in body weight throughout the process (Fig. S3b). In a more permissive model,

the SW620 subcutaneous colorectal tumor model, M1-N3E2M potently suppressed tumor growth, while its parental virus, M1-GFP, inhibited tumor growth to a lesser extent (Fig.3g, h). There was no significant difference in body weight among the three groups (Fig. S3c). At the end of the experiment, the mice were sacrificed, and the tumors were excised. The size of the tumors in the M1-N3E2M group was clearly smaller than that of the tumors in the control and M1-GFP groups, and the tumor weight was lower (Fig. S4). We further explore the antitumor activity using immune-competent mice with subcutaneous tumors derived from CT26 and Hepa1-6, two widely used mouse models for colorectal and liver cancer, respectively. The results showed that M1-N3E2M significantly inhibited tumor growth, whereas M1-GFP did not (Fig. 3i–l). Importantly, all mice remained asymptomatic, and no significant differences in body weight were observed during the observation period (Fig. S3d, e). These results demonstrate that M1-N3E2M has stronger therapeutic potential than M1-GFP for suppressing tumor growth without harming normal tissues.

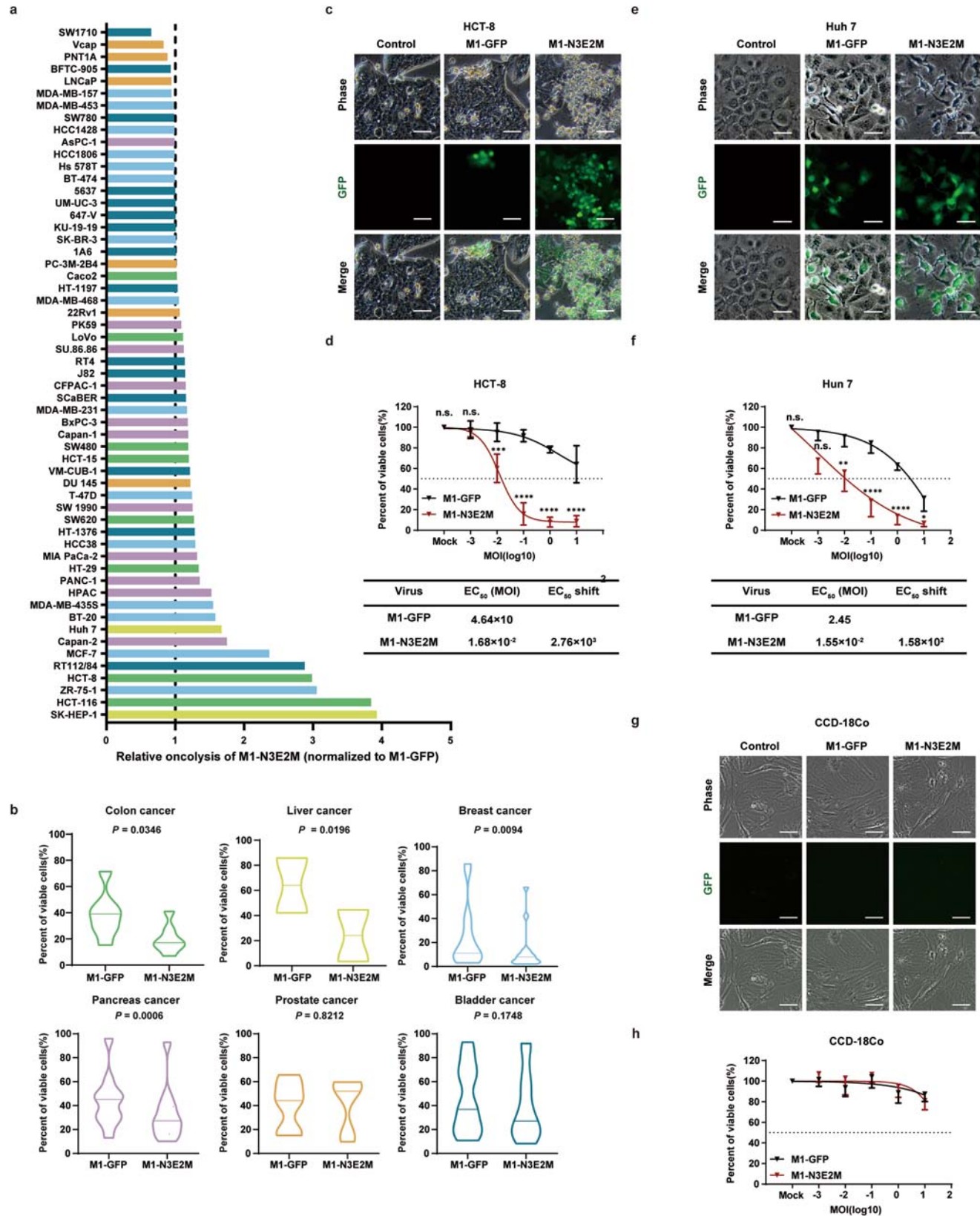

To provide more clinically relevant evidence, we established an ex vivo model with surgical tumor samples from colorectal carcinoma patients. Tumor tissues isolated from six colorectal carcinoma patients were divided into small pieces, which were assayed for viability after exposure to M1-GFP, M1-N3E2M, or vehicle for 72 h. The sensitivity of tumor tissues to M1 virus differed greatly, possibly owing to the heterogeneity of the tumors. In tumor tissues with high sensitivity to M1-GFP (cytotoxicity>10%), the oncolytic effect of the two viruses was similar. However, in tumor tissues refractory to M1-GFP (cytotoxicity<10%), the oncolytic effect of M1-N3E2M was enhanced (Fig. 3m), suggesting its potential for treating colorectal carcinoma.

**Fig. 2 | The oncolytic effect of M1-N3E2M was improved in a variety of tumor cells, and it did not cause a CPE in normal cell line. a** The viability of 57 human cell lines were evaluated by an MTT assay 72 h after infection. Liver cancer is shown in yellow, colon cancer in green, prostate cancer in orange, pancreas cancer in purple, bladder cancer in blue and breast cancer in light blue. **b** The oncolytic effects of M1-GFP and M1-N3E2M were analyzed by Two-tailed paired t test and $P$ values are indicated. The data are shown as violin plots with the box limits at minima and maxima and center line at median. (colon cancer, $n = 8$; liver cancer, $n = 2$; breast cancer, $n = 15$; pancreas cancer, $n = 11$; prostate cancer, $n = 6$; bladder cancer, $n = 15$). **c, e** HCT-8 and Huh-7 cells were infected with M1 viruses at an MOI of 0.1. Representative images of $n = 3$. Scale bars, 50 μm. **d, f** The viability of HCT-8 and Huh-7 cells was evaluated by an MTT assay. EC50 shift was calculated by nonlinear regression. Statistical significance was calculated using Two-way ANOVA with Sidak's multiple comparisons test relative to M1-GFP. Adjusted $P$ values are: **d** MOI (Mock), $P > 0.9999$; MOI ($-3$), $P = 0.9991$; MOI ($-2$), $P = 0.0008$; MOI ($-1$ to 1), $P < 0.0001$; **f** MOI (Mock), $P > 0.9999$; MOI ($-3$), $P = 0.0528$; MOI ($-2$), $P = 0.0028$; MOI ($-1$ and 0), $P < 0.0001$; MOI (1), $P = 0.0489$. **g** CCD-18Co cells were infected with M1-GFP and M1-N3E2M at an MOI of 10. Representative images of $n = 3$. Scale bars, 50 μm. **h** The viability of CCD-18Co cells was evaluated by an MTT assay. EC50 shift was calculated by nonlinear regression. n.s.: no significance, $*P < 0.05$, $**P < 0.01$, $***P < 0.001$, $****P < 0.0001$. Data points represent mean % viability relative to vehicle ± SD, for $n = 3$ biological replicates. Source data are provided as a Source Data file.

## Mutations in E2 and nsP3 synergistically facilitate the oncolysis of M1 virus

To determine the contributions of these two mutations to the improved oncolytic efficacy of M1-N3E2M, we generated M1-E2M and M1-NS3M, which contained the K4N mutation in E2 and the M358L mutation in nsP3, respectively. We examined the oncolytic effect of M1 viruses in HCT-116 cells by the MTT method and found that the oncolytic effect of the M1 viruses with individual mutations was stronger than that of M1-GFP but weaker than that of M1-N3E2M, indicating that both mutations partially contributed to the enhanced oncolysis of M1 virus and that they functioned synergistically (Fig. 4a). Individually, the M358L mutation in nsP3 resulted in stronger enhancement than the K4N mutation in E2 (Fig. 4a). The effects on viral replication and yield, as indicated by viral protein expression and viral titers, were consistent with those on cell viability. First, observations by fluorescence microscopy showed a slight increase in the replication of M1-E2M but a considerable increase in that of M1-NS3M. The M1 virus with both mutations exhibited greater replication than either of the individual mutants (Fig. 4b). Second, quantitative analysis of the abundance of viral structural protein E1 confirmed what we observed by fluorescence microscopy (Fig. S5). Last but not least, we compared the viral production capacity of the M1 variants in HCT-116 cells and found that all M1 variants produced more progeny viruses than the parental virus. A slight increase in viral yield was observed for M1-E2M, while the yield of M1-NS3M increased substantially, and both mutations were required for the full boost of viral replication (Fig. 4c).

To further dissect the specific role of either mutation in potentiating viral replication, we performed a plaque formation experiment. HCT-116 cells were washed with PBS to remove unbound virus particles 1 h after infection, and the virus particles that had entered the cells could then replicate to form plaques[21]. The viral entry speed determines the number of plaques formed, and the amount of local viral replication affects the size of the plaques. Compared with M1-GFP, infection with M1-E2M resulted in significantly more plaques, and infection with M1-NS3M resulted in larger plaques. When both mutations were present, the number and size of plaques increased concomitantly (Fig. 4d-f). These findings suggest that the K4N mutation in E2 may accelerate the entry of M1 virus, while the M358L mutation in nsP3 may promote viral replication. Collectively, these results indicate that these two mutations synergistically potentiate the oncolytic effect of M1 virus via complementary mechanisms.

## The K4N mutation in E2 accelerates the entry of M1 virus

Previously, we obtained the cryo-EM structures of M1 virus and the M1-receptor complex, illustrating that the envelope protein E2 is on the surface of virus particles, and plays a critical role in receptor binding[22]. To delineate the mechanism by which the K4N mutation in E2 accelerates the entry of M1 virus, we first performed a plaque formation-based entry kinetics assay and found that M1-E2M virus formed significantly more plaques as early as 5 min after nfection, confirming that it can indeed enter tumor cells faster than the parental virus (Fig. 5a, b).

Consistent with the finding, M1-E2M cells had an increased attachment ability to tumor cells (Fig. 5c).

Given that Mxra8 has been identified as the receptor for M1 virus[22], we sought to determine whether M1 virus with the K4N mutation in E2 has a higher binding affinity for Mxra8. A pull-down assay was exploited to evaluate the binding ability between M1 viruses and the extracellular domain of the MXRA8 protein. Viral particles were first incubated with MXRA8-His protein and then immunoprecipitated with anti-His Sepharose beads. A greater amount of virus, represented by envelope protein E1, was observed in the M1-E2M group, suggesting that the interaction between M1-E2M and the MXRA8 receptor protein was stronger (Fig. 5d). Biolayer interferometry (BLI) further confirmed that M1-E2M virus particles bound to the MXRA8 receptor with significantly higher affinity (Fig. 5e). We next sought to figure out whether the higher receptor binding affinity contributes to the more rapid entry of M1-E2M. In HeLa cells, a cell model with extremely low expression of MXRA8[23], we observed low infection rates for both M1-GFP and M1-E2M. Ectopic expression of MXRA8 in HeLa cells resulted in a much greater increase in the infection rate for M1-E2M than for the parental virus (Fig. 5f). Conversely, knockout of MXRA8 in Hs 578 T cells, a cell model with a high level of MXRA8 expression, reduced the difference in the infection rate between M1-GFP and M1-E2M (Fig. 5g). Taken together, these findings indicate that the K4N mutation in E2 endows M1 virus with a higher binding affinity for the receptor MXRA8 and accelerates the attachment and entry of M1 virus.

## The M358L mutation in nsP3 promotes the replication of M1 virus

Next, we attempted to unveil the underlying mechanism by which the M358L mutation in nsP3 promotes viral replication. First, flow cytometric analysis showed not only an increased infection rate after M1-NS3M infection (Fig. 6a) but also an elevated GFP expression level in infected cells (Fig. 6b), indicating that M1-NS3M had superior replication capacity. Second, analysis of entry kinetics showed that the number of plaques formed by M1-NS3M was similar to that for M1-GFP (Fig. 6c), suggesting that the M358L mutation in nsP3 may not affect viral entry.

The M358L mutation is located in the hypervariable domain (HVD) of nsP3, which is reported to interact with a variety of host proteins to facilitate viral replication[24]. To further clarify the molecular mechanism of the M358L mutation in nsP3, an interaction proteomics approach was used to identify the host proteins that interact with the wild-type and mutated nsP3 proteins (Fig. 6d). Some of the well-studied proteins that interact with nsP3 of other alphaviruses, including Ras GTPase-activating protein-binding protein 1 (G3BP-1), G3BP-2, and SH3 domain-containing kinase-binding protein 1 (SH3KBP1)[25–28], were also found in our analysis, supporting the reliability of this experiment. In addition, 59 proteins specifically interacted with wild-type nsP3, and 22 proteins specifically interacted with nsP3-M358L (Fig. 6e, Table S1 and S2)[29]. While the wild-type nsP3-specific interactors were mainly enriched in mRNA processing pathways (Fig. 6f), the mutated nsP3-specific interactors were enriched in virus-related pathways (Fig. 6g),

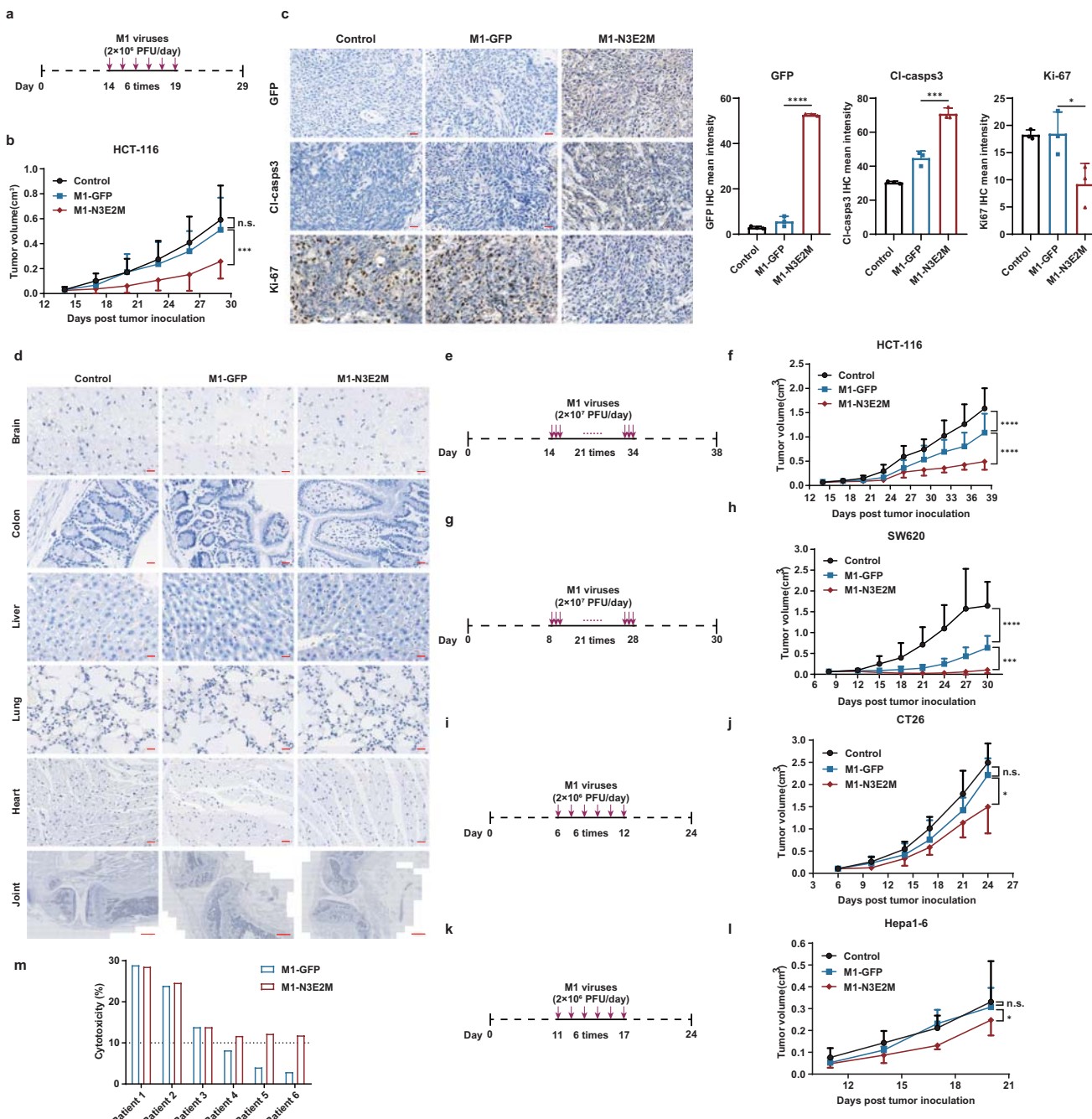

**Fig. 3 | The oncolytic effect of M1-N3E2M was potentiated in vivo and ex vivo.**
**a**, **b** HCT-116 xenografts were treated intravenously with vehicle, M1-GFP or M1-N3E2M for 6 consecutive days. $n = 7$ mice per group. **c** Tumor tissues were analyzed by immunohistochemistry for GFP, cleaved caspase-3, and Ki-67. Representative images of $n = 3$. Scale bars, 20 μm. Quantification of GFP, cleaved caspase-3 and Ki-67, $n = 3$ biological replicates. Statistical significance was calculated using One-way ANOVA with Sidak's multiple comparisons test relative to M1-GFP. Adjusted $P$ values are: GFP $P < 0.0001$, Cl-casps-3 $P = 0.0002$, Ki-67 $P = 0.0376$. **d** Normal tissues from the brain, colon, liver, lungs, heart (Scale bars, 20 μm) and joints (Scale bars, 500 μm). were analyzed by immunohistochemistry for GFP. Representative images of $n = 3$. **e–h** HCT-116 and SW620 xenografts were treated intravenously for 21 consecutive days. In HCT-116 xenograft model, $n = 7$ mice per group; in SW620 xenograft model, $n = 10$ mice per group. **i–l** CT26 xenografts in BALB/c mice and HEPA1-6 xenografts in C57BL/6 mice were treated intravenously

for 6 consecutive days. In CT26 xenograft model, $n = 5$ mice per group; in HEPA1-6 xenograft model: Control $n = 7$ mice, M1-GFP $n = 9$ mice, M1-N3E2M $n = 8$ mice. **m** Colorectal tumor tissues from six patients were treated with M1-GFP or M1-N3E2M ($1 \times 10^7$ PFUs) for 72 hours, and cell viability was assessed. One graph bar represents the mean cytotoxicity % relative to vehicle of one tumor sample. Statistical significance of tumor volume was calculated using Two-way ANOVA with Tukey's multiple comparisons test. Adjusted $P$ values are: **b** M1-GFP vs. control, $P = 0.4464$; M1-N3E2M vs. M1-GFP, $P = 0.0005$; **f** M1-GFP vs. control, $P < 0.0001$; M1-N3E2M vs. M1-GFP, $P < 0.0001$; **h** M1-GFP vs. control, $P < 0.0001$; M1-N3E2M vs. M1-GFP, $P = 0.0006$; **j** M1-GFP vs. control, $P = 0.0673$; M1-N3E2M vs. M1-GFP, $P = 0.0127$; **l** M1-GFP vs. control, $P = 0.6654$; M1-N3E2M vs. M1-GFP, $P = 0.0176$. n.s.: no significance, *$P < 0.05$, **$P < 0.01$, ***$P < 0.001$, ****$P < 0.0001$. Data are shown as mean ± SD. Source data are provided as a Source Data file.

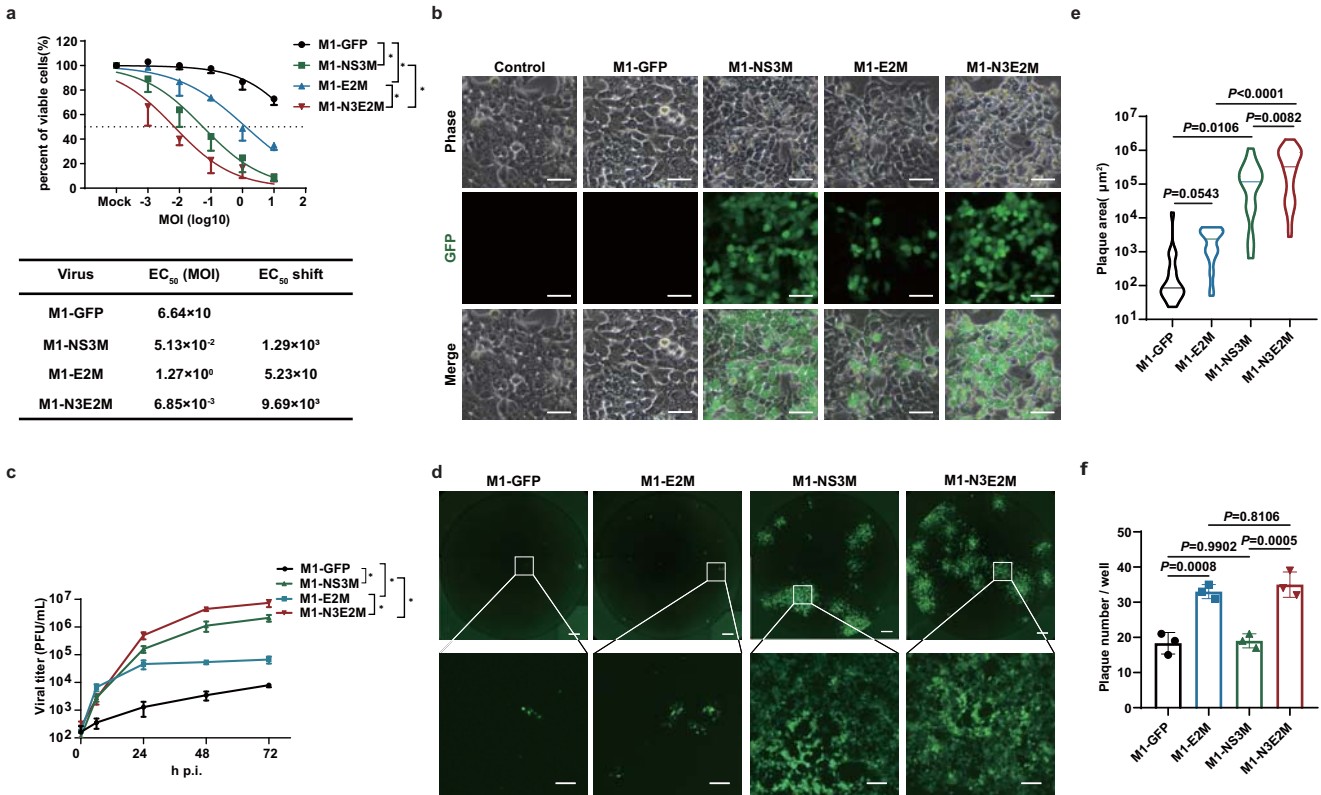

**Fig. 4 | The oncolytic effect of M1-N3E2M was synergistically enhanced by the M358L and K4N mutations. a** Cell viability was evaluated by an MTT assay. EC50 shift was calculated by nonlinear regression and the EC50 values were used for statistical analysis by unpaired *t*-test with Welch's correction, M1-NS3M vs. M1-GFP, $P = 0.0467$; M1-E2M vs. M1-GFP, $P = 0.0481$; M1-N3E2M vs. M1-NS3M, $P = 0.0345$; M1-N3E2M vs. M1-E2M, $P = 0.0237$. Data points represent mean % viability relative to vehicle ± SD, for $n = 3$ biological replicates. **b** HCT-116 cells were infected with M1 viruses at an MOI of 0.1 and imaged 48 h after infection. Representative images of $n = 3$. Scale bars, 50 μm. **c** The viral titer was determined by a TCID50 assay. We performed a statistical analysis of the final virus production using unpaired *t*-test with Welch's correction, M1-NS3M vs. M1-GFP, $P = 0.0240$; M1-E2M vs. M1-GFP, $P = 0.0327$; M1-N3E2M vs. M1-NS3M, $P = 0.0451$; M1-N3E2M vs. M1-E2M, $P = 0.0284$. Data points represent mean viral titer ± SD, for $n = 3$ biological

replicates. **d** Monolayer HCT-116 cells were infected with M1 viruses at an MOI of 0.1. The medium was replaced with semisolid medium 1 h after infection. Representative images of $n = 3$. Scale bars, 500 μm. The scale bars in the magnified images represent 100 μm. **e** Quantification of the plaque area in (**d**). Statistical significance was calculated using unpaired *t*-test with Welch's correction and $P$ values are indicated. The data are shown as violin plots with the box limits at minima and maxima and center line at median (M1-GFP $n = 21$, M1-E2M $n = 30$, M1-NS3M $n = 17$, M1-N3E2M $n = 32$). **f** The plaques in (**d**) were counted. Statistical significance was calculated using One-way ANOVA with Tukey's multiple comparisons test and $P$ values are indicated. Graph bars represent mean plaque number per well ± SD, for n = 3 biological replicates. n.s.: no significance, *$P < 0.05$, **$P < 0.01$, ***$P < 0.001$, ****$P < 0.0001$. Source data are provided as a Source Data file.

suggesting an important role of the M358L mutation in nsP3 during M1 infection[30]. Among the mutated nsP3-specific interactors, PKR, one of the well-known antiviral proteins in the defense against RNA viruses, attracted our attention. We further validated the selective interaction between PKR and mutated but not wild-type nsP3 by co-immunoprecipitation (Fig. 6h). To uncover the biological impacts of this interaction on M1 virus replication, we first examined the activation of PKR. As early as 4 h after infection, the phosphorylation of PKR was significantly upregulated by M1-GFP but not by M1-NS3M. At 24 h after infection, the phosphorylation of PKR induced by M1-NS3M remained much weaker than that induced by M1-GFP (Fig. 6i). These results suggest that interaction with nsP3-M358L may inhibit the activation of PKR. In light of the observation that phosphorylated PKR can mediate the expression of genes in the type I interferon (IFN) pathway, we further checked the expression and activation of STAT1, one of the indispensable transcription factors in the type I IFN pathway. Between the two viruses, neither the expression nor the phosphorylation level of STAT1 was changed at 4 h after infection. At 24 h after infection, however, M1-GFP resulted in significant increases in the total STAT1 and p-STAT1 levels, while M1-NS3M did not (Fig. 6j). Accordingly, the expression of multiple IFN-stimulated genes (ISGs), which are

transcriptionally regulated by STAT1, was significantly less induced by M1-NS3M than by M1-GFP (Fig. 6k, l).

To demonstrate that the enhanced replication of M1-NS3M is dependent on PKR binding, we generated PKR knockdown HCT-116 cell lines to compare virus growth. The suppression of PKR increased the infection rate of M1-GFP to equivalent level as M1-N3M (Fig. 6m–o). This finding suggests that the nsP3 M358L mutation inhibits the antiviral response and improves viral replication by inhibiting PKR. Even when PKR was knockdown, the infection of M1-N3E2M was still higher than that of M1-GFP and M1-N3M, further confirming the contribution of the E2 K4N mutation in enhancing receptor binding (Fig. S6).

To address safety issues, we further investigated whether M1-NS3M affects the activation of the PKR-STAT1 pathway in normal cells. CCD-18Co normal colon fibroblasts were used, and we found that both M1-GFP and M1-NS3M significantly activated PKR and STAT1 at 24 h after infection, resulting in intense inhibition of viral replication (Fig. 6p, q). The above results illustrate that nsP3-M358L interacts with PKR and suppresses its activation, thus endowing M1-NS3M with the ability to evade antiviral responses by inhibiting the activation of the PKR-STAT1-IFN signaling pathway and enhancing virus replication in tumor cells.

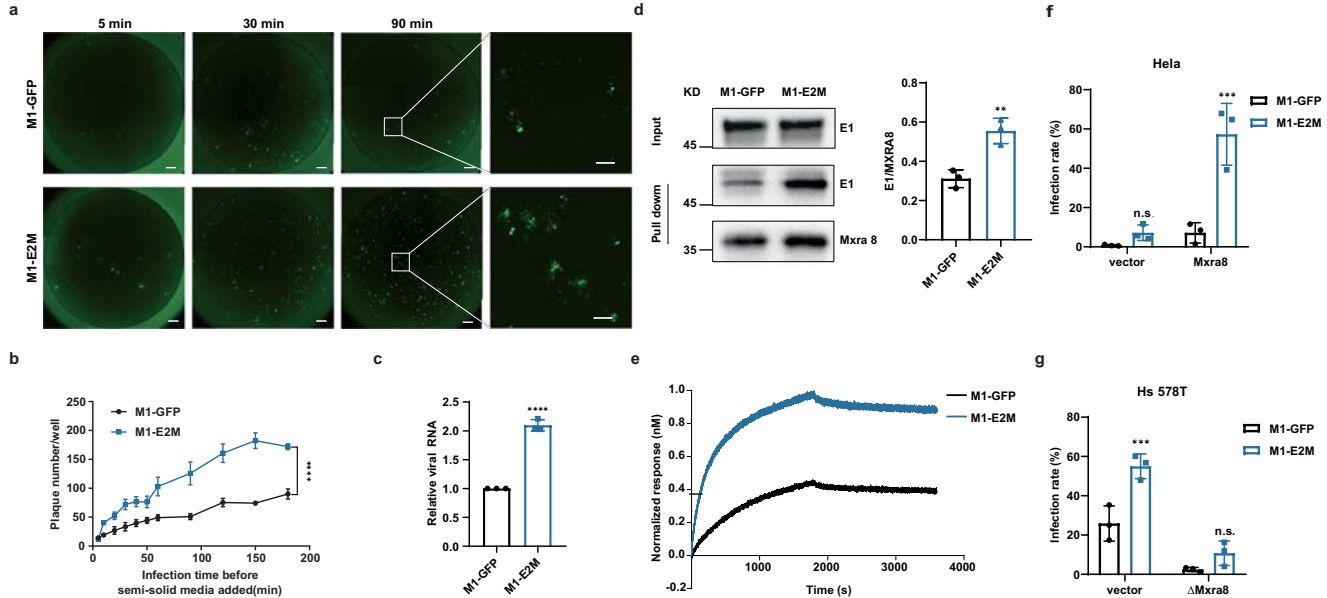

**Fig. 5 | The K4N mutation in E2 improved the attachment and entry of M1 virus.**
**a** A modified plaque formation assay was performed and cells were imaged with a fluorescence microscope 48 hours after infection (MOI = 0.5). Scale bars, 500 µm. The scale bars in the magnified images represent 100 µm. **b** Plaques were counted with a fluorescence microscope 48 hours after infection. Data points represent mean plaque number per well ± SD, for $n$ = 3 biological replicates. $P$ < 0.0001 was determined by Two-way ANOVA relative to M1-GFP. **c** HCT-116 cells were incubated with M1-GFP and M1-E2M at 4 °C for 1 hours. Viral RNA was quantified by qRT-PCR and presented as mean ± SD, for $n$ = 3 biological replicates. $P$ < 0.0001 was calculated with Two-tailed unpaired t-test. **d** Western blot analysis of M1-GFP and M1-E2M incubated with MXRA8-His bound to His-Tag Mouse mAb Sepharose Beads. Precipitated viral particles were detected using an anti-E1 mAb (left). Quantification

of E1 expression is shown (right). Graph bars represent mean densitometry of E1 normalized to the densitometry of MXRA8 ± SD, for $n$ = 3 biological replicates. $P$ = 0.0060 was calculated with Two-tailed unpaired t-test. **e** Time course of the binding between M1 viral particles and the MXRA8 protein, as determined via BLI. **f, g** HeLa, HeLa-Mxra8, Hs 578 T and Hs 578T-ΔMxra8 cells were treated with M1-GFP or M1-E2M. The infection rate was determined by flow cytometry 48 hours after infection. Graph bars represent mean infection rate % ± SD, for $n$ = 3 biological replicates. Statistical significance was calculated using Two-way ANOVA with Sidak's multiple comparisons test relative to M1-GFP. Adjusted $P$ values are: **f** vector $P$ = 0.6199; Mxra8 $P$ = 0.0002; **g** vector, $P$ = 0.0010; ΔMxra8 $P$ = 0.2713. n.s.: no significance, **$P$ < 0.01, ***$P$ < 0.001, ****$P$ < 0.0001. Source data are provided as a Source Data file.

## M1-N3E2M is safe in nonhuman primates

Safety evaluation is one of the essential steps in the development of novel OVs. In previous studies, we verified the safety of wild-type M1 virus in rodent and nonhuman primate models[10,11]. Our data also indicated that M1-N3E2M retained the tumor-selective properties of the parental M1 virus, thus ensuring its high safety, which was supported by the results in our mouse models (Fig. S3). For further confirmation of safety and in consideration of future clinical translation, we performed a comprehensive safety assessment in *Macaca fascicularis*. The administration schedule is shown in Fig. 7a, and the injected dose of M1-N3E2M was 3.29×10⁹ CCID50 per animal. During the test period, all animals were under clinical observation, and no dead or dying animals were observed, nor was drug-related toxicity. No adverse reactions at the administration site were reported (Table S3). Loss of body weight was not observed (Fig. 7b). There were no significant differences in the animals' vital signs, including body temperature, heart rate and blood pressure (Fig. 7c). During the experiment, no obvious abnormalities were found during the ophthalmological examination of all animals (Table S4).

Except for transient increases in individual animals in the control group, the concentrations of liver enzymes (ALT and AST) and albumin did not change significantly after virus administration (Fig. 7d). The urinalysis and the concentrations measurement of serum creatinine and urea showed that renal function was not compromised (Table S5, Fig. 7d).

In hematological analyses, the counts of leukocytes, lymphocytes, neutrophils, and monocytes showed slight fluctuations but were not greatly different from those before administration (Fig. 7e). There was no overt abnormality in the coagulation function of the animals in each group (Fig. 7f). During the experiment, there was no visible systemic

toxicity except for transient elevation of IL-6 and HsCRP levels in the control group (Fig. 7g). Even though the levels of complement C3 and C4 decreased on Day 44 after M1-N3E2M injection, they eventually recovered to the basal levels (Fig. 7h). Neutralizing IgG antibodies were detectable at low levels 20 days after M1-N3E2M administration (Table S6). Overall, these results indicate that the M1-N3E2M virus was safe in nonhuman primates, providing strong evidence of the safety profile to support its clinical translation.

## Discussion

In this study, we demonstrated that the efficacy of OV M1 was potent by experimental evolution in refractory tumor cells. The evolved M1 virus efficiently killed a broader spectrum of tumor cells in vitro and suppressed tumor growth in mouse subcutaneous cancer models. Mechanistically, we demonstrated that the entry of M1 virus was accelerated via an increase in its binding to the Mxra8 receptor and that the evolved M1 virus overcame the antiviral response by inhibiting the activity of PKR and STAT1 in tumor cells. Importantly, it remained harmless to normal human cell lines, normal mouse tissues, and nonhuman primates.

Due to the inadequate understanding of viral gene function, manipulations to attenuate virulence limit viral replication in many cases, reducing the oncolytic potential of OVs[3]. Directed evolution is an approach to obtain virus strains with desirable phenotypes when the function of genes is unclear. Therefore, OVs with better safety and more specificity for malignant cells can be produced by directed evolution. Using directed evolution, we can overcome the differences in the oncolytic effect on tumor cells caused by tumor heterogeneity. In the present study, directed evolution enabled M1 virus to acquire adaptive mutations in refractory tumor cells and greatly broadened

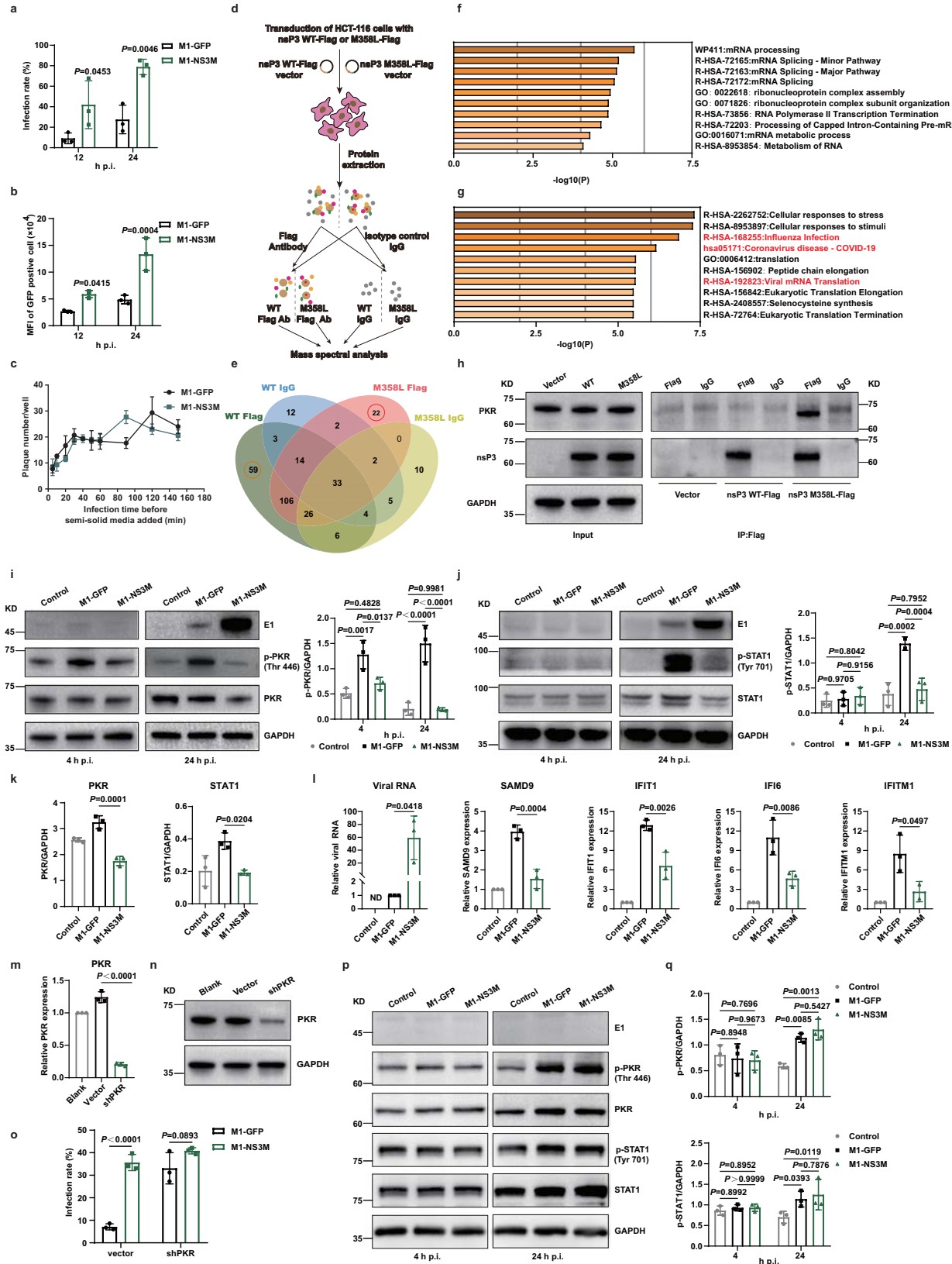

the antitumor spectrum of M1 virus. This suggests that a variety of optimized OVs can be generated by directed evolution in different tumor cells, providing the possibility for personalized and precise treatment of cancer patients.

The K4N mutation in the evolved M1 accelerated the attachment and entry of M1 virus by enhancing the interaction of E2 with the receptor Mxra8, thereby increasing the oncolytic effect by 52-fold. The acceleration of virus entry increased the number of infected cells but this increase was not sufficient to support viral replication in malignant cells. Meanwhile, the M358L mutation in nsP3 enhanced the replication of M1 virus via evasion of PKR-mediated antiviral responses. The inhibition of the intracellular antiviral pathway allowed the virus to

**Fig. 6 | The M358L mutation in nsP3 inhibited the activation of PKR and STAT1, further inhibiting IFN-mediated antiviral responses. a** HCT-116 cells were infected with M1-GFP and M1-NS3M (MOI = 10), and the infection rate was determined by flow cytometry. **b** The expression of GFP in infected cells in (**a**) was detected. MFI, mean fluorescence intensity. **c** A modified plaque formation assay was performed and plaques were counted with a fluorescence microscope 48 hours after infection. **d** Schematic of the process for identifying interactions with nsP3 WT and nsP3 M358L. **e** This Venn diagram shows the host proteins that interact with nsP3 WT and nsP3 M358L. **f, g** Bar graph of enriched terms across the 59 nsP3 WT interactors and 22 nsP3 M358L interactors, colored by p values. The top 10 enriched pathways are shown. See also Table S1 and S2. **h** Coimmunoprecipation was conducted with an anti-Flag antibody or isotype control IgG prior to immunoblot analysis with anti-nsP3 and anti-PKR antibodies. Representative images of *n* = 3. **i, j** HCT-116 cells were treated with control, M1-GFP or M1-NS3M for 4 and 24 hours (MOI = 10), and the levels of proteins in the cell lysates was examined by Western blotting (left). Quantification of p-PKR and p-STAT1 (right). **k** Quantification of PKR and STAT1 expression in (**i, j**). **l** The transcript levels of ISGs and viral RNA were quantified by qRT–PCR after 24 h infection with control, M1-GFP or M1-NS3M (MOI = 1). (**m-o**) qRT-PCR (**m**) and western blotting (**n**) were used to evaluate the shRNAs knockdown efficiency of PKR. PKR knockdown cells were infected with M1-GFP and M1-NS3M (MOI = 1) and the infection rate was determined by flow cytometry (**o**). **p, q** Proteins were detected by Western blotting in CCD-18Co cells after infection with M1-GFP or M1-NS3M (MOI = 1). Quantification of p-PKR and p-STAT1. Statistical significance was calculated using Two-way ANOVA with Sidak's multiple comparisons test (**a, b, i, j, o, q**) or One-way ANOVA with Tukey's multiple comparisons test (**k–l**), and adjusted *P* values are indicated. Data are shown as mean ± SD from three biological replicates. Source data are provided as a Source Data file.

replicate in large numbers, resulting in an order-of-magnitude increase in viral yield. Therefore, the oncolytic effect of M1 virus boosted dramatically, by more than 6400- to 9600-fold, under the synergistic effect of the two mutations. Changes in any step in the viral life cycle may affect the oncolytic effect of M1 virus, and accelerating different steps at the same time may be a new strategy to potentiate the oncolytic effects of OVs.

Direct oncolysis of OVs is limited mainly due to antiviral innate immunity[31,32]. PKR, a key antiviral protein, is activated by a variety of cellular stress signals, especially the classical activator, double-stranded RNA. Activated PKR blocks viral protein synthesis by phosphorylating eukaryotic translation initiation factor 2 alpha (EIF2a)[33,34]. It has been reported that viruses have developed multiple mechanisms to inhibit PKR to evade innate immune responses. Nonstructural protein 5 A (NS5A) of hepatitis C and the E3L protein of vaccinia virus inhibit the activation of PKR by direct interaction[35,36], while nonstructural protein 1 (NS1) of influenza virus inhibits the activation of PKR by RNA sequestration[37,38]. There is no report that alphaviruses inhibit the PKR response in the existing studies. PKR has not been identified among the numerous interacting proteins of alphavirus nsP3[24]. The mechanism by which mutant nsP3 inhibits PKR remains unclear and needs further investigation. However, M1 virus with the M358L mutation was replication-incompetent in normal cells due to the inability to evade antiviral immunity.

The safety of wild-type M1 virus has been elucidated in previous reports[11,19]. We validated the safety profile of M1-N3E2M virus in cellular experiments, animal models and nonhuman primates. Our results demonstrate the safety of multiple repeated high-dose intravenous injections of M1-N3E2M virus in nonhuman primates.

## Methods

### Ethics statement
The mouse experiments were performed under the protocol approved by the Animal Ethics and Welfare Committee of Sun Yat-sen University (no. 2016-114) and the Laboratory Animals Ethics Committee of Lani Scientific (Guang Zhou) Co., Ltd. (no. G2022024). The nonhuman primates used in this study were approved by the Institute's Animal Management and Use Committee (IACUC, no. ACU18-1162), and the experiments were conducted according to the Guide for the Care and Use of Laboratory Animals, 8th Edition. All samples were collected with the patients' written informed consent and approved by the ethical review board of the Six Affiliated Hospital of Sun Yat-sen University (no. L2019ZSLYEC-144).

### Cell culture
Cell line details are shown in supplementary table S8, including the names and source of the cell lines, their STR profiling status, and their mycoplasma testing results. HeLa-Mxra8 were generated by transducing HeLa cells with the Mxra8-expressing lentiviral vector, and Hs 578T-ΔMxra8 were generated by using CRISPR-Cas9 gene editing technology[22]. Cells were cultured in Dulbecco's modified Eagle's medium (DMEM) (Corning) or RPMI 1640 medium (Corning) supplemented with 10% (vol/vol) fetal calf serum (Gibco) and 1% penicillin/streptomycin (Thermo Fisher Scientific). All cells were incubated at 37 °C in a 5% $CO_2$ incubator. Images of cells were captured with NIS-Elements viewer 4.20.

### Virus production
M1 virus was produced in Vero cells, which were cultured in VP-SFM (Thermo Fisher Scientific) supplemented with 10% (vol/vol) MEM-NEAA and GlutaMAX (Thermo Fisher Scientific). When approximately 80% of the Vero cells were infected and showed a marked CPE, the supernatant was collected, centrifuged at 2000×g and 4 °C for 10 min, and stored at −80 °C.

### Virus titer
BHK-21 cells were seeded in 96-well plates at 3000 cells per well and incubated for 24 h. The virus-containing supernatant was serially diluted in DMEM and was then added to the BHK-21 cells. Seventy-two hours after infection, GFP expression and the CPE were evaluated under a fluorescence microscope. The virus titer was calculated using Spearman-Karber method and converted to PFU.

### Serial passaging of M1 virus
In the serial passage experiment, 3×10^5 HCT-116 cells were inoculated into each 35 mm culture dish, and 1.5 mL of medium was added. Recombinant M1-GFP was added at an MOI of 10, and the cell status and virus replication were observed daily. When the GFP signal no longer increased, the supernatant was collected, centrifuged at 2000×g and 4 °C for 10 min, and stored at −80 °C. This batch of viruses was named P1. After collecting the supernatant, 1 mL of TRIzol (Life Technologies) was added to the dish and stored at −80 °C for RNA extraction. The P1 virus was then added to new fully adherent HCT-116 cells in a volume of 100-300 μL, and the next round of culture was carried out. When the replication of a certain generation of virus had increased significantly, the supernatant was collected, and a 10 μL aliquot of the virus was used to infect HCT-116 cells.

### Cell viability and infection rate assay
Cells were seeded in 24-well plates at 20000 cells per well in 500 μL of medium. After 72 h of infection with M1 viruses at different MOIs, 3-(4,5-dimethylthiazol-2-yl)−2,5-diphenyltetrazolium bromide (MTT) was added (1 mg/mL), and incubated at 37 °C for 2–4 h. The MTT-containing medium was removed, and the formazan crystals were dissolved in 500 μL of DMSO. The optical absorbance was measured at 570 nm using a microplate reader (Biotek Synergy H1), and data were collected with Gen 5. The infection rate was determined by flow cytometry and CytExpert 2.4 was used to collect data.

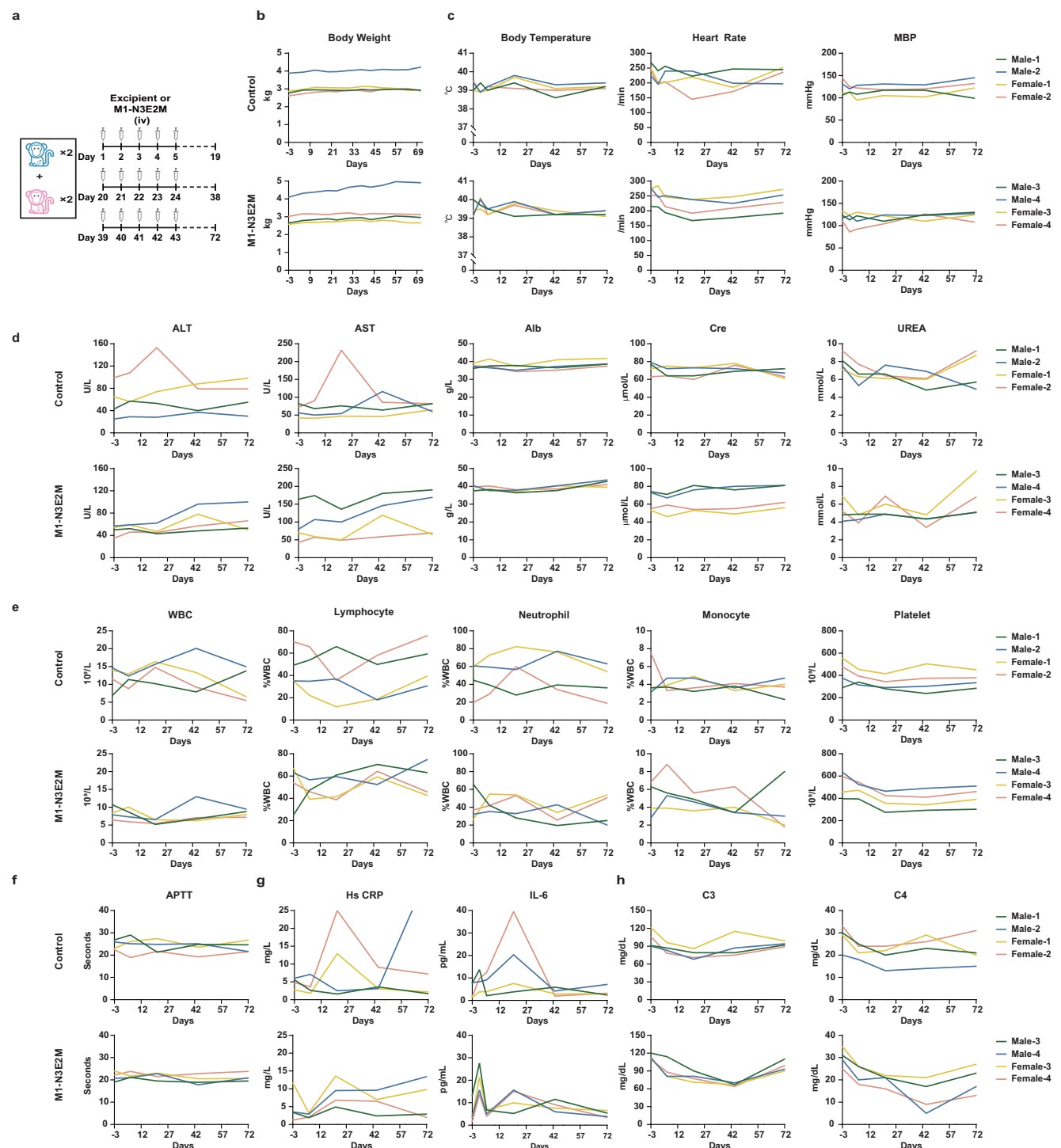

**Fig. 7 | M1-N3E2M is well tolerated in nonhuman primates. a** Timeline of the administration schedule. iv, intravenously. **b** Animals were weighed weekly after treatment initiation. **c** Animal body temperature, electrocardiographic and blood pressure measurements were conducted on D-3, D1, D5, D20, D43 and D71. MBP, mean blood pressure. **d** Blood was collected from a subcutaneous vein of the hind limbs of animals on D-3, D1, D6, D21, D44 and D72 for analysis of serum biochemical parameters. ALT, alanine aminotransferase. AST, aspartate aminotransferase. Alb, albumin. Cre, creatinine. **e** Blood was collected from a subcutaneous vein of the hind limbs of animals on D-3, D1, D6, D21, D44 and D72 for analysis of hematological parameters. WBC, white blood cell. **f** Analysis of blood coagulation function on D-3, D6, D21, D44 and D72. APTT, activated partial thromboplastin time. **g** Cytokine and serum C-reactive protein detection. **h** Complement detection on D-3, D1, D6, D21, D44 and D72. See also Table S3-S6. Source data are provided as a Source Data file.

## Construction of M1-NS3M, M1-E2M, and M1-N3E2M

The M1-GFP plasmid vector described previously in Ref. 20 was used as the backbone to construct the M1-NS3M, M1-E2M, and M1-N3E2M plasmids. To generate M1-NS3M, two PCR fragments were amplified from the M1-GFP plasmid using primers 5173 F and 5230 R or 5298 F and 7104 R. The nsP3 mutant fragment was obtained by fusing the two PCR fragments using the outer primers 5173 F and 7104 R. The mutant fragment was digested with SpeI and SwaI (Thermo Scientific) and inserted into the M1-GFP plasmid digested with the same restriction enzymes. To generate M1-E2M, two PCR fragments were amplified

from the M1-GFP plasmid using primers 9420 F and 9510 R or 9486 F and 10521 R. The fragments were fused to obtain the E2 mutant fragment using the outer primers 9420 F and 10521 R. Then, the mutant fragment was digested with XhoI and ApaI (Thermo Scientific) and inserted into the M1-GFP plasmid digested with the same restriction enzymes. Furthermore, the nsP3 mutant fragment was inserted into the M1-E2M plasmid to obtain the M1-N3E2M plasmid. All the plasmids were validated by DNA sequencing (Thermo Fisher Scientific). Viral RNAs were in vitro transcribed from linearized plasmids using an SP6 RiboMAX Large-Scale RNA Production System (Promega) and were then transfected into Vero cells with Lipofectamine MessengerMAX (Thermo Fisher). The supernatant was collected when approximately 80% of the Vero cells were infected and showed a marked CPE. The primers (Thermo Fisher Scientific) included the following (5′ to 3′):

5173 F TTGACCAGACCGTCCCGTCACTAGTAAGTCCCAGAAAGTACATACAGCA

5230 R ACTTCCAGGGTTTCGTAGGTCGT

5298 F ACGACCTACGAAACCCTGGAAGT

7104 R CTGACTTCATCATAGCACCGAATTTAAATCTTGTACCGGTAGGTAGATGCACACTCGT

9420 FCGGGCTACTACGACCTGCTCGAGGCCACGATGACGTGTAACAACAGTGCACGCC

9510 R  TTGTAGACATTGAAGTGGTTCGTCACACTGCGACGGTGGCGTGCACTGTTGTTACACG

9486 F TGACGAACCACTTAATGTCTACAA

10521 R  GGTTTGCCTTCAGTTGTCAGCTGGGCCCACAAGCGCACGGGTGGG.

## Genome sequencing
Total RNA was extracted from samples frozen during serial passaging, and reverse transcription was performed to synthesize cDNA according to the manufacturer's instructions for the GoScript Reverse Transcription System (Promega). PCR was performed with Q5 High-Fidelity 2× Master Mix (NEB) and ten pairs of amplification primers (Thermo Fisher). The sequences of the primers are listed in Table S7. The PCR fragment amplified with primers 9936 and 11696 was inserted into a vector with a ClonExpress Entry One Step Cloning Kit (Vazyme). All PCR fragments were sequenced (Thermo Scientific) and then aligned with the M1-GFP sequence with Lasergene software 7.1.0.

## Animal models
The maximal tumor burden is 3000 mm$^3$ permitted by ethics committee. Four- to six-week-old female BALB/c-nude mice (Beijing Vital River Laboratory Animal Technology Co., Ltd.) were housed in a pathogen-free room in the experimental animal center of Sun Yat-sen University. HCT-116 and SW620 cells were subcutaneously inoculated into the rear flanks of BALB/c-nude mice at a density of $5 \times 10^6$ cells/mouse. Six- to eight-week-old female BALB/c mice and C57BL/6 mice (Gempharmatech Co., Ltd.) was used for CT26 xenograft subcutaneous models and HEPA1-6 xenograft subcutaneous models, respectively. BALB/c mice and C57BL/6 mice were housed in pathogen-free room in Lani Scientific (Guang Zhou) Co., Ltd.. Mice were housed at an ambient temperature of 22-24 °C, humidity-controlled environment at 40%-70% under a 12-h light/dark cycle with ad libitum access to water and food. When the tumor volume was approximately 50 mm$^3$, the mice were randomly divided into groups, and viruses were administered by tail vein injection for 6 or 21 consecutive days. The administered dosage was $2 \times 10^6$ or $2 \times 10^7$ PFU/day of M1 viruses or an equal volume of vehicle. Tumor lengths and widths were measured every three days to calculate the tumor volumes (length×width$^2$/2). When the tumor volume was up to 3000 mm$^3$, it was considered as the humane endpoint. After treatment, tumor tissues and normal organs, such as the liver, heart, brain, and lungs, were harvested and fixed with 4% paraformaldehyde for subsequent analysis. The number of mice is indicated in the figure legend.

## Immunohistochemistry assay
Paraffin tumor sections were baked in a 60 °C oven for 2 h, dewaxed in xylene, hydrated in decreasing concentrations of ethanol, and soaked in 0.3% H$_2$O$_2$-methanol for 30 min. Then, the tumor sections were immersed in sodium citrate buffer for antigen retrieval and blocked with standard 1.5% goat serum for 30 min. The proliferation, apoptosis, and replication of M1 viruses were evaluated with monoclonal anti-Ki-67 (#9449 s, Cell Signaling Technology; 1:800), anticleaved caspase-3 (#9664 s, Cell Signaling Technology; 1:2000) and monoclonal anti-GFP (#2956 s, Cell Signaling Technology; 1:400) antibodies or isotype control at 4 °C overnight. After washing, the sections were treated with appropriate secondary antibodies at room temperature. Finally, the sections were visualized with DAB Staining Solution and counterstained with hematoxylin.

## Ex vivo assays
We used tissue culture endpoint staining computer image analysis (TECIA) to evaluate the oncolytic effect of M1-N3E2M ex vivo[39]. Since no significant difference related to sex was found in the M1 antitumor study, we did not conduct gender analysis. Primary colorectal carcinoma tissues derived from tumors surgically resected from patients, providing by the Sixth Affiliated Hospital, Sun Yat-sen University, was maintained in medium supplemented with high concentrations of penicillin and streptomycin. The tissue samples were aseptically cut into blocks of approximately 1 mm$^3$ and cultured in 24-well plates for 24 h with 1 ml of DMEM containing 15% (vol/vol) FBS and 10% (vol/vol) penicillin/streptomycin. The colorectal carcinoma tissues were exposed to M1-GFP or M1-N3E2M virus ($1 \times 10^7$ PFU) for 72 h. Cell viability was evaluated by an MTT assay.

## Plaque formation assay
A modified plaque formation assay was performed to measure the entry kinetics of the M1 variants. Briefly, 30,000 HCT-116 cells per well were seeded in 96-well plates. The cells were grown overnight and the culture medium was then replaced with fresh medium containing M1 viruses. The infection medium was aspirated at 5, 10, 20, 30, 40, 50, 60, 90, 120 and 180 min after infection. Following aspiration of the infection medium, the cells were washed three times with PBS to remove external or unbound virus particles, and a semisolid overlay medium containing 0.75% agarose was added to cover the cells. Forty-eight hours after infection, the plaques (GFP-positive) were counted with a fluorescence microscope. Viral entry kinetic curves were generated by normalizing the relative plaque number to the incubation time to evaluate the entry speed of the M1 variants. The plaque areas were calculated with Image J 1.46r software.

## CoIP and MS analysis
The cDNA sequences of wild-type nsP3 and nsP3 M358L were chemically synthesized (GENEWIZ, Suzhou, China) with a 3×flag tag. The two gene fragments were inserted into the pEZ-Lv206 vector, which were used to generate lentiviral particles for transferring wild-type and mutant nsP3 protein to mammalian cells (GeneCopoeia). In the presence of polybrene, HCT-116 cells were infected by the lentiviral particles (MOI = 1). Cells cultured 72 h post-transduction and selected against with Puromycin(0.5 µg/ml) (Thermo Fisher Scientific), and the expression of mCherry was detected with a fluorescence microscope. Then, total protein was extracted. The protein concentration was quantified with a BCA protein assay kit (Thermo Fisher) and adjusted to a concentration of 2 mg/mL for the following coimmunoprecipitation experiments. One microgram of a mouse monoclonal anti-FLAG antibody (#F3165, Sigma) or normal mouse IgG (#5415, Cell Signaling Technology) was added to each sample and incubated in a swing-type incubator overnight at 4 °C. Protein A/G (Bimake) beads were used to precipitate the interacting protein

complexes, which were subsequently analyzed by mass spectrometry (Shanghai Applied Protein Technology).

## Western blot analysis

Cells were collected and homogenized with 150 μL of M-PER Mammalian Protein Extraction Reagent (Thermo Scientific) and were then incubated for 15 min on ice. The supernatant was isolated by centrifugation at $12,000 \times g$ for 10 min at 4 °C. The concentration of total protein in each sample was quantified by the BCA method (Thermo Fisher). Equal amounts of protein were separated by SDS-PAGE and then transferred to membranes (Millipore). The membranes were blocked with 5% milk (Millipore) for 1 h and incubated with mouse monoclonal anti-E1 (produced by Beijing Protein Innovation; 1:1000), rabbit monoclonal anti-STAT1 (#14994, Cell Signaling Technology; 1:1000), rabbit monoclonal antiphospho-STAT1 (#9167, Cell Signaling Technology; 1:1000), rabbit monoclonal anti-PKR (#12297, Cell Signaling Technology; 1:1000) and rabbit monoclonal antiphospho-PKR (#ab32036, Abcam; 1:1000) antibodies in blocking solution at 4 °C overnight. After three washes with Tris-buffered saline (TBS), the membranes were incubated with the corresponding secondary antibody at room temperature for 1 h. To confirm equal protein loading, the membranes were also probed with an antibody against GAPDH (#5174, Cell Signaling Technology; 1:1000). The target proteins were detected using a ChemiDoc XRS+ System (Bio-Rad). Bio-Rad Image lab 5.2.1 software was used to measure the relative band densities for quantification of protein expression. Source data are provided as a Source Data file.

## Attachment assay

The attachment assay was performed in HCT-116 cells seeded in 6-well plates. The cells, M1-GFP and M1-N3E2M were prechilled at 4 °C for 15 min prior to coincubation for 1 h at 4 °C. The HCT-116 cells were washed with PBS five times to remove unbound viruses, and 1 mL of TRIzol was then added to extract RNA.

## qRT-PCR

Total RNA was extracted from frozen samples and reverse transcribed into cDNA (Thermo Fisher). Quantitative PCR was performed with SuperReal PreMix SYBR Green (TIANGEN) using an Applied Biosystems 7500 Fast Real-Time PCR System (Life Technologies). Relative cDNA levels were calculated by the comparative CT (cycle threshold) method. The PCR primers (Thermo Fisher Scientific) included the following (5′ to 3′):

 M1 NS1 forward primer GTTCCAACAGGCGTCACCATC
 M1 NS1 reverse primer ACACATTCTTGTCTAGCACAGTCC
 β-actin forward primer GATCATTGCTCCTCCTGAGC
 β-actin reverse primer ACTCCTGCTTGCTGATCCAC

## Pull down assay

M1-GFP and M1-E2M ($1\times10^7$ PFU) virus particles, 1 μg of His-Mxra8 (Sino Biological), and His-Tag Mouse mAb Sepharose Beads (Cell Signaling Technology) were incubated overnight at 4 °C in TBS containing 10 mM $CaCl_2$. After centrifugation at $14,000 \times g$ for 30 s, the supernatant was removed, and the beads were washed three times with TBS containing 10 mM $CaCl_2$ and 0.05% Tween 20. The beads were resuspended in 50 μL of loading buffer and boiled for 5 min, and proteins eluted into the supernatant were detected by Western blotting. The E1 protein was the target protein for detection, and Mxra8 was used as a control.

## Biolayer interferometry

M1 viruses propagated in Vero cells were purified by centrifugation at $5,000 \times g$ for 15 min to remove cellular debris, and the supernatant was collected and filtered through a 0.22 μm filter. Then, the supernatant was collected by centrifugation at $32,000 \times g$ for 1 h in PBS. The titers of M1 viruses were determined by RT-qPCR as copy numbers per microgram of RNA. MXRA8 proteins were mixed with biotin at a ratio of 1:1 at room temperature for 30 min, and unreacted biotin was removed with a desalting column (Genemore). Biotinylated MXRA8 was loaded onto streptavidin biosensors (ForteBio) at 5 μg/mL for 5 min in PBS (pH 7.4) containing 0.005% Tween 20, and M1 viruses were then added at 1 μM in the same buffer. Association was monitored over a 30 min period, followed by a 30 min dissociation period at 25 °C. The experiment was performed on an Octet RED96e instrument following the manufacturer's protocol.

## Safety evaluation

Eight pathogen-free *cynomolgus macaques* (four males and four females) were reared and handled at JOINN Laboratories (China) Co., Ltd., according to the Guide for the Care and Use of Laboratory Animals, 8th Edition. Based on the animal body weights determined before administration (D-3), the animals were randomly divided into 2 groups by sex (2/sex/group) using the Provantis 9.4.3.0 system and administered an aqueous solution of excipients or M1-N3E2M ($4.7\times10^9$ PFU). All animals were administered a total of 15 doses intravenously on D1-D5, D20-D24, and D39-D43. We chose the dose based on the following considerations:(1) Previous study. We had intravenously administered three rounds of wild-type M1 virus ($1.4\times10^9$ PFU/dose, six doses/round) in cynomolgus macaques for safety evaluation. The results support a high safety profile of M1 virus. (2) Clinical trial. Oncolytic virus M1 is undergoing a clinical trial, and the dosage used is $1\times10^9$ CCID50 once daily on days 1–5, every 28-day cycle. We increased the dosage and shortened the time period slightly ($2\times10^9$ PFU once daily on days 1–5, every 19-day cycle) for safety evaluation of the mutant virus to provide flexibility for future design of clinical trials.

During the experiment, the animals were subjected to clinical observation, body weight measurement, body temperature measurement, electrocardiography, blood pressure measurement, ophthalmological examination, blood cell count determination, coagulation function determination, blood biochemical analysis, urinalysis, C-reactive protein measurement, complement measurement, antibody, and cytokine measurement.

## Statistical analysis

Data are expressed as the means ± SDs. Statistical analysis of the data was completed using SPSS 18.0 and GraphPad Prism 8. The unpaired $t$-test was used to compare two sets of data, and one-way analysis of variance (ANOVA) was used to compare the mean responses among the treatment groups. Tumor growth was statistically analyzed by Two-way ANOVA with Tukey's multiple comparisons test. $P$ values of $<0.05$ were considered indicate statistically significant differences. $*P < 0.05$, $**P < 0.01$, $***P < 0.001$, $****P < 0.0001$. The corresponding statistical methods and $P$ values are mentioned in each figure or figure legend.

## Reporting summary

Further information on research design is available in the Nature Portfolio Reporting Summary linked to this article.

# Data availability

All data relevant to the study are included in the article or uploaded as supplementary information. The virus genome data are available from the Nucleotide database of National Center for Biotechnology Information under accession number OP683724, OP683725, OP683726, and OP683727. The raw data of protein interaction are available from PRIDE database (https://www.ebi.ac.uk/pride/archive/projects/PXD037429). The remaining data are available within the Article, Supplementary Information or Source Data file. Source data are provided with this paper.

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

## Acknowledgements

We thank Ziqing Lin and Suzhen Zhang (Guangzhou Virotech Pharmaceutical Co., Ltd, #3 Lanyue Road, Science Park, Guangzhou 510663, China.) for technical assistance; we thank Bernard Roizman, Sc.D. for theoretical guidance. Funding: This project was funded by National Key R&D Program of China (2021YFA0909800, Y. Lin), National Natural Science Foundation of China (81973347, Y. Lin and 82173837, J.L.), China Postdoctoral Science Foundation (2019TQ0395, L.G.), The Fundamental Research Funds for the Central Universities, Sun Yat-sen University (22ykqb12, Y. Lin) and Guangdong Basic and Applied Basic Research Foundation (2020A1515110153, Z.D. and 2020A1515110907, J.C.).

## Author contributions

Conceptualization: L.G., Y. Lin, and G.Y. Methodology: L.G., Y.Liu, D.S., X.J., Y.G., and Q.Z. Investigation: L.G., C.H., Y.Liu, X.C., D.S., R.S., and Z.L. Formal analysis: L.G., Y. Lin., and Z.D. Supervision: G.Y., J.H., W.Z. C.J., and J.L. Writing – Original Draft: L.G. Writing – Review & Editing: Y. Lin., J.L. and T.Z.

## Competing interests

The authors declare no competing interests.
