## [Peer Review File · Nature Communications]

Directed natural evolution generates a next-generation oncolytic virus with a high potency and safety profileReviewers' Comments:

Reviewer #1:

Remarks to the Author:

This is an interesting paper that seeks to improve the oncolytic activity of an alphavirus based oncolytic called M1. The authors use a natural selection strategy by sequentially passaging on tumour cells that are intrinsically resistant to M1. This strategy has been used before to improve the oncolytic activity of adenoviruses and rhabdoviruses. The authors find a new variant that they claim is 6500X more effective than the parental strain and apparently improves virus output by 10K fold. They show the virus can kill some cancer cell lines more effectively in vitro and yet retains its selectivity by not killing normal cell lines (cultures) Additionally it appears safe in rodent and NHP models. They identify the two genes that are mutated to provide the increased virus output. While there is a great deal of data, I find it is often over-interpreted. For instance:

(1) the authors claim to see differential association of the mutant virus proteins with PKR and impact on phospho PKR in figure 6 but the blots look identical to me.

(2) the ability of the virus to have so significantly improved virus output and killing capacity is not reflected in the improved therapeutic activity shown in mouse models or in the patient explants (figure 3).

(3) if the activity of the novel virus is truly related to PKR binding than the M1 and N3E2M viruses should grow equivalently in PKR -/- cells.

(4) The changes in the interferon response proteins the authors examined are again very modest compared with the reported improvements in virus yield and killing (figure 6)

Reviewer #2:

Remarks to the Author:

This is a very interesting study by Dr. Guo, Liang, Lin and colleagues. There is considerable interest in the oncolytic virotherapy arena to find novel vectors with selectivity to tumor cells vs. normal cells that is retained and with higher infectivity, potency and viral particle production. The study team developed a recombinant M1 alphavirus (positive strand RNA virus) that is an arbovirus through serial passaging based evolution in HCT-116 cell line. Two proteins were found to be mutated in multiple passages - E2 at K4N and nsP3 at M358L. The authors showed that there was more pronounced toxicity in vitro compared to the wild type vector across a proportion of cell lines, particularly those of gastrointestinal origin. Additionally, there was more infectivity and viral particle production in these cases with the mutant virus. In vivo studies were done in immune deficient models using cell lines from the in vitro experiments with intravenous administration to show that efficacy was retained with no significant increase in toxicity. The mechanistic aspects for the activity were delineated to be due to higher receptor binding for E2 mutant and decreased PKR/STAT1 signaling for the nsP3 mutant. A pilot safety study was done in *Macaca fascicularis* which did not yield any concerning safety signals. Overall, the work was well laid out and interesting.

One major limitation of the work is the absence of any in vivo studies done in immune competent models. This would be critical to more thoroughly address the efficacy of any oncolytic virus, especially if PKR/STAT1 signaling is felt to be a mechanistic underpinning of the differential activity.

Reviewer #3:

Remarks to the Author:

This is an interesting study by Guo et. al., that investigates the anti-tumor potential of a novel variant of the oncolytic virus M1. 2 mutations that enhance virus replication were identified by experimental evolution on colon cancer cells. Insights into the mechanisms by which these mutations enhance the virus as well as safety data are provided.

While some results are interesting, this approach has been used in the past. Some mechanistic data is lacking to convincingly establish the effect of the different mutations. Additional experiments would be needed to properly assess the tumor-selectivity of the virus.

General comments:

1. In many figures, stats are missing. Ex: 1C, H, 2D, F
2. The introduction, as well as the discussion, are very long and should be shortened. Repetitions of the intro, as well as unnecessary details are found in the discussion, which should be more concise.
3. The study would benefit from a more comprehensive assessment of the effect of the mutant virus in normal cells. Should a higher MOI be used? Is there a "normal" cell line that would allow for infection? If not, plasmids could be used to study the interaction of the viral protein with PKR in normal cells. As is, this part is not convincing.

Specific comments:

1. Fig 2A could be color-coded for different cancer types. This would facilitate the analysis.
2. Fig. 2B: figure legends would probably be too big, but as is, it is impossible to determine which cell line is which. I find this figure pretty weak. Maybe simple bar charts with error bars would show the differences better. As is, it seems that most differences, when any, are very small. Bar charts would also allow for a statistical analysis, which could help convince the reader. Overall, the data found in Figs 2A and B should be consolidated and represented in a more comprehensive and convincing way.
3. The conclusion that digestive system tumors are more sensitive to the virus variant is somehow misleading. The cell lines tested are mostly of these types and not all of them are more sensitive. The virus does not seem to be specifically more sensitive for digestive system tumors. The text and conclusions should be reworded to reflect that the virus is more potent in other tumor types, not just in specific cancers.
4. Fig. 3. What is the rationale for 21 daily injections? This seems like a lot and no justification is provided.
5. Figs 3C, H and K should be moved to supplemental. Or at least H and K since it is very repetitive.
6. Figs. 3J, L and M show the same thing. L and M should be moved to supplemental.
7. Unless I am mistaken, 3N had replicates. Why no error bars? Also, why 10% as the cutoff? This affects the interpretation and is not justified.
8. Fig. 4C should be moved to supplemental.
9. Fig 4D: virus output does not increase over time. Does virus production stop after 24h? Please explain this.
10. Fig. 5D is missing a control input.
11. Fig. 6H, why is there a signal for PKR in all conditions (including control). And why is there an NSP3 signal in the Ig control pulldown?
12. "Neither the expression nor the phosphorylation level of STAT1 was changed at 4 hours after infection". Between the 2 viruses? Or with control? There is a signal for both viruses.
13. Fig. 7A. How was the dose selected? Based on previous studies? Please explain in the text.

Reviewer #4:

Remarks to the Author:

Review NCOMMS-22-39722-T

The manuscript "Directed natural evolution generates a next-generation oncolytic virus with a high potency and safety profile for gastrointestinal cancers" describes the generation of oncolytic viruses

(OVs) as drug treatment against cancer, in this case gastrointestinal cancers. These OVs have been made more potent in virulence, infectivity, and oncolytic capacity by serial passage through a colorectal carcinoma cell line. The generated OVs showed effectiveness in a much broader spectrum of tumor cells than the original M1 variant while they were unharmed to normal cell lines. The functional effectiveness of the adapted OVs have been extensively tested in various in vivo models while safety was evaluated in non-human primates.

The paper describes extensive experiments showing the increased infectivity and oncolytic effects of the passaged OVs and their mechanisms behind. The results are clear and convincing and do show the potency that directed natural evolution could be a generalizable approach for developing next-generation OVs.

Some questions:

- The therapeutic potential of M1-N3E2M was tested against the HCT-116 tumor in Nude mice with different frequency of the OVs administered (Fig 3A vs 3F). Obviously, the effect of the low frequency of administration of OVs (3A) gave similar results as the more frequent administration. Wouldn't it make sense to further decrease the number of OVs administration or is there a minimum number of OVs administration required to see an effect?
- There was a significant decrease in tumor sizes after M1-N3E2M treatment in HCT-116 mice (Fig 3B and G) and in SW620 mice (3J) while the animals did not show any weight loss (3C, H and K). How do the authors explain that there was a reduction of tumor weight from 1.5 – 2 gr to approx. 0.2 gr (Fig 3M) while the weight of the animals remains around 18-20 grams (Fig 3K) and did not change over time. A reduction of 2 grams tumor weight in animals of approx. 20 grams is a reduction of 10%! Maybe the authors can give changes in body weight as a percentage rather than the actual body weights.
- Why is the frequency of OVs (3 x 5 injections) in macaques different that those in Nude mice studies? Is this administration strategy based on potential clinical trials? Please explain.

Point-by-point response to the reviewers' comments

We thank the reviewers for their constructive and insightful feedback on our manuscript. Below, we provide a detailed point-by-point response to each comment. The original comments are quoted in blue and our responses are in black. The changes we have made in the revised manuscript are highlighted in yellow.

Reviewer #1:

1. The authors claim to see differential association of the mutant virus proteins with PKR and impact on phospho PKR in figure 6 but the blots look identical to me.

Response: We agree with the reviewer and postponed that the slightly differential PKR association shown in the original Fig. 6H may be owing to the low level of nsP3 expression in our transient transfection experiments. To test this hypothesis, we generated HCT-116 cell lines that stably expressing wild-type or mutant nsP3 protein using lentivirus transduction and performed PKR interaction analysis. Our results show that nsP3 with M358L mutation strongly interacts with PKR, while the wild-type does not. We have replaced the Fig. 6H with this new result in the revised manuscript.

Revised Fig. 6H. HCT-116 cell lines with stable expression of wild-type and mutant nsP3 protein were cultured in the presence of puromycin. Co-immunoprecipitation was conducted with an anti-Flag antibody or isotype control IgG prior to immunoblot analysis with anti-nsP3 and anti-PKR antibodies.

Accordingly, we have included the methods of the above experiment in the MATERIALS AND METHODS section in line 525-530 of page 24 in the revised manuscript as follows, “The two gene fragments were inserted into the pEF1alpha-

IRES-AcGFP1 vector with the restriction enzymes EcoRI and XmaI (Thermo Scientific). In the presence of Lipofectamine 3000 (Thermo Fisher), the vectors were transfected into HCT-116 cells for 48 hours, and total protein was then extracted” was replaced by “The two gene fragments were inserted into the pEZ-Lv206 vector, which were used to generate lentiviral particles for transferring wild-type and mutant nsP3 protein to mammalian cells (GeneCopoeia). In the presence of polybrene, HCT-116 cells were infected by the lentiviral particles (MOI=1). Cells cultured 72 h post-transduction and selected against with Puromycin(0.5µg/ml) (Thermo Fisher Scientific), and the expression of mCherry was detected with a fluorescence microscope. Then, total protein was extracted.”.

We also measured the phospho-PKR levels in HCT-116 cells infected with a higher dose (10 MOI) of virus, enabling mutated nsP3 to be expressed in most cells, thereby suppressing PKR by interaction. Our results showed that M1-GFP significantly upregulated PKR phosphorylation, whereas M1-NS3M did not induce PKR activation (revised Fig. 6I). Consistently, M1-GFP induced STAT1 phosphorylation, whereas M1-NS3M did not (revised Fig. 6J). PKR and STAT1 protein expression levels were also increased by M1-GFP but not by M1-NS3M (revised Fig. 6K). We have updated these data in the revised Fig. 6I-K.

Revised Fig. 6I-K. (I) HCT-116 cells were treated with control, M1-GFP or M1-NS3M for 4 and 24 hours (MOI=10), and the levels of proteins in the cell lysates was examined by Western blotting with anti-E1, anti-p-PKR (Thr 446) and anti-PKR antibodies (left). Quantification of p-PKR expression (right). $n = 3$; mean \pm SD; **** $P < 0.0001$, ** $P < 0.01$, * $P < 0.05$. (J) HCT-116 cells were treated with control, M1-GFP or M1-NS3M for 4 and 24 hours (MOI=10), and the levels of proteins in the cell lysates was examined by Western blotting with anti-E1, anti-p-STAT1 (Tyr 701) and anti-STAT1 antibodies (left). Quantification of p-STAT1 expression (right). $n = 3$; mean \pm SD; *** $P < 0.001$; ns, not significant. (K) Quantification of PKR and STAT1 expression in (I) (K). $n = 3$; mean \pm SD; * $P < 0.05$, *** $P < 0.001$.

2. The ability of the virus to have so significantly improved virus output and killing capacity is not reflected in the improved therapeutic activity shown in mouse models or in the patient explants (figure 3).

Response: We concur with the reviewer that while the improved in vitro killing effect

is noteworthy, the increase in ex vivo or in vivo therapeutic activity is less significant. This may be attributed to the antiviral effects of innate and adaptive immune responses observed in mouse models or patient explants. Other oncolytic viruses, including vesicular stomatitis virus and adenovirus, have reported similar outcomes, where the killing ability and viral production in vitro were notably increased, but the effect in vivo was relatively moderate^{1, 2}. These observations imply that enhancing infection and replication in cancer cells alone is insufficient to dramatically improve the in vivo therapeutic effect of oncolytic viruses, and that the tumor microenvironment should also be considered.

3. If the activity of the novel virus is truly related to PKR binding then the M1 and N3E2M viruses should grow equivalently in PKR -/- cells.

Response: We appreciate the reviewer's suggestion and have generated PKR knockdown HCT-116 cell lines to compare the growth of different viruses. The suppression of PKR increased the infection rate of M1-GFP to equivalent level as M1-N3M (Revised Fig. 6O). This finding suggests that the nsP3 M358L mutation inhibits the antiviral response and improves viral replication by inhibiting PKR. Even when PKR was knockdown, the infection of M1-N3E2M was still higher than that of M1-GFP and M1-N3M (Revised Fig. S5), further confirming the contribution of the E2 K4N mutation in enhancing receptor binding. We have included these additional findings in the revised Fig. 6M-O and Fig. S5, and have added the text to line 284-290 of page 14 in the revised manuscript.

Revised Fig. 6M-O. Short hairpin PKR (shPKR) lentiviral particles encoding mCherry reporter and the corresponding shControl lentiviral particles were purchased

from OBiO Technology Corp., Ltd. (Shanghai, China). HCT-116 cells were transfected with the lentiviral particles and were selected using puromycin (500 ng/ml) (Thermo Fisher Scientific) at 72h post-transduction. qRT-PCR (**M**) and western blotting (**N**) were used to evaluate the shRNAs knockdown efficiency when almost all the cells were mCherry positive. PKR knockdown cells were infected with M1-GFP and M1-NS3M at an MOI of 1 and the infection rate was determined by flow cytometry (**O**).

Revised Fig. S5. (A) PKR knockdown cells were infected with M1-GFP, M1-NS3M and M1-N3E2M at an MOI of 1, and imaged with a fluorescence microscope 24 h after infection. Scale bars, 50 μ m. (B) The infection rate was determined by flow cytometry.

4. The changes in the interferon response proteins the authors examined are again very modest compared with the reported improvements in virus yield and killing (figure 6)

Response: We concur with the reviewer's observation that the changes in expression levels of the interferon-stimulated genes (ISGs) tested in our study appear to be modest, in comparison to the observed improvements in viral yield and killing. However, it is important to note that different ISGs can exert antiviral effects on distinct stages of the virus life cycle. For instance, SAMD9 can restrict viral replication³, IFIT1 can inhibit viral protein translation⁴, IFI6 can play a critical role in immunomodulation and antiapoptotic processes, as well as the entry process of various viruses⁵, and IFITM can block viral entry by inhibiting viral fusion⁶. Moreover, these ISGs can function synergistically. Thus, the moderate but significant inhibition of multiple ISGs caused by M1-NS3M may eventually lead to a noticeable enhancement in viral yield and killing.

Reviewer #2

One major limitation of the work is the absence of any *in vivo* studies done in immune competent models. This would be critical to more thoroughly address the efficacy of any oncolytic virus, especially if PKR/STAT1 signaling is felt to be a mechanistic underpinning of the differential activity.

Response: We appreciate the reviewer's suggestion. We conducted additional *in vivo* experiments using immune competent mice with subcutaneous tumors derived from CT26 and Hepa1-6, two widely used mouse models for colorectal and liver cancer, respectively. To this end, we injected 2×10^6 PFU/day of M1 viruses or an equal volume of vehicle intravenously for 6 consecutive days. The results showed that M1-N3E2M significantly inhibited tumor growth, whereas M1-GFP did not. Importantly, all mice remained asymptomatic, and no significant differences in body weight were observed during the observation period. These findings have been incorporated in the revised Fig. 3I-L and Fig. S2D&E, and have added the text to line 180-185 of page 9 in the revised manuscript.

Revised Fig. 3I-L. (I, J) CT26 xenografts in BALB/c mice were treated intravenously with vehicle, M1-GFP or M1-N3E2M for 6 consecutive days, and tumor volumes (J) were measured. $n=5$; mean \pm SD; $*P<0.05$; *ns*, not significant. (K, L) Hepa1-6 xenografts in C57BL/6 mice were treated intravenously with vehicle, M1-GFP or M1-N3E2M for 6 consecutive days. Tumor volumes (L) were measured every three days. Control: $n=7$; M1-GFP: $n=9$; M1-N3E2M: $n=8$; mean \pm SD; $*P<0.05$; *ns*, not significant.

Revised Fig. S2D, E. (D) Relative body weights of mice in Fig. 3I were measured. $n=5$; mean \pm SD; *ns*, not significant. (E) Relative body weights of mice in Fig. 3K were measured every three days. Control: $n=7$; M1-GFP: $n=9$; M1-N3E2M: $n=8$; *ns*, not significant.

Reviewer #3:

1. In many figures, stats are missing. Ex: 1C, H, 2D, F

Response: We have added the stats in these figures according to the reviewer's comment in the revised manuscript and have checked the other figures for any missing stats.

Fig. 1C

Fig. 1H

Fig. 2D

Fig.2F

Fig. 4A

(Fig.1C) Cell viability was evaluated by an MTT assay 72 h after cells were infected with serial dilutions of M1-GFP, P3, P8 and P10. The EC₅₀ values were used for statistical analysis and EC₅₀ shift was calculated by nonlinear regression. n=3; mean±SD. (Fig. 1H) Cell viability was evaluated 72 h after cells were infected with serial dilutions of M1-GFP and M1-N3E2M. EC₅₀ shift was calculated by nonlinear regression. n=3; mean±SD. *P<0.05; ***P<0.001; ****P<0.0001; ns, not significant.

(**Fig. 2D**) The viability of HCT-8 cells was evaluated by an MTT assay 72 h after infection with serial dilutions of M1-GFP and M1-N3E2M. EC50 shift was calculated by nonlinear regression. n=3; mean±SD. (**Fig. 2F**) Huh-7 cells were treated with M1-GFP and M1-N3E2M (MOI=0.001, 0.01, 0.1, 1 and 10). EC50 shift was calculated by nonlinear regression. n=3; mean±SD. (**Fig. 4A**) Cell viability was evaluated by an MTT assay 72 h after HCT-116 cells were infected with serial dilutions of M1-GFP, M1-NS3M, M1-E2M, or M1-N3E2M. The EC50 values were used for statistical analysis and EC50 shift was calculated by nonlinear regression. n=3; mean±SD.

2. The introduction, as well as the discussion, are very long and should be shortened. Repetitions of the intro, as well as unnecessary details are found in the discussion, which should be more concise.

Response: We appreciate the reviewer's suggestion. We have revised the introduction and discussion accordingly. Changes highlighted in yellow in the revised manuscript. Line 58-60, 81-82, 315-316, 324-325, 335-342, 363-365 and 368-380 were deleted. Line 360-363 were added.

3. The study would benefit from a more comprehensive assessment of the effect of the mutant virus in normal cells. Should a higher MOI be used? Is there a "normal" cell line that would allow for infection? If not, plasmids could be used to study the interaction of the viral protein with PKR in normal cells. As is, this part is not convincing.

Response: To address the reviewer's suggestion for a more comprehensive study with normal cells, we first evaluated the viability of normal cells infected with wild-type or mutant virus at higher MOIs. We found that none of the three normal cell lines tested exhibited significant cytotoxicity after exposure to either virus at 100 MOI, which is 10-fold higher than the highest MOI used in cancer cells. These data are shown in supplementary Fig. 1 and line 151-153 of page 8 in the revised manuscript. Additionally, we generated NCM 460 normal colon cell lines stably expressing wild-type or mutant nsP3 protein by lentivirus transduction to examine the PKR interaction.

We confirmed that PKR interacted with mutated but not wild-type nsP3 protein in normal cells as well. We have incorporated this result in supplementary Fig. 6, and line 299-302 of page 14 in the revised manuscript.

Fig. S1. Killing effects of M1 viruses on normal cell lines. CCD-18Co, NCM 460 and L-02 cells were treated with M1-GFP and M1-N3E2M (MOI=20, 40, 60, 80 and 100). EC50 shift was calculated by nonlinear regression. n = 3; mean±SD.

Fig. S6. NCM 460 cell lines with stable expression of wild-type and mutant nsP3 protein were cultured in the presence of puromycin. Coimmunoprecipitation was conducted with an anti-Flag antibody or isotype control IgG prior to immunoblot analysis with anti-nsP3 and anti-PKR antibodies.

Specific comments:

1. Fig 2A could be color-coded for different cancer types. This would facilitate the analysis.

Response: In response to the reviewer’s suggestion, we have color-coded Fig. 2A to distinguish different cancer types. Liver cancer is yellow, colon cancer is green, prostate cancer is orange, pancreas cancer is purple, bladder cancer is blue and breast cancer is light blue (line 796-797 of page 36 in the revised manuscript).

Revised Fig. 2A Seventy-three human cell lines were infected with M1-GFP and M1-N3E2M at an MOI of 10 and evaluated by an MTT assay 72 h after infection. Liver cancer is yellow, colon cancer is green, prostate cancer is orange, pancreas cancer is purple, bladder cancer is blue and breast cancer is light blue.

2. Fig 2B: figure legends would probably be too big, but as is, it is impossible to determine which cell line is which. I find this figure pretty weak. Maybe simple bar charts with error bars would show the differences better. As is, it seems that most differences, when any, are very small. Bar charts would also allow for a statistical analysis, which could help convince the reader. Overall, the data found in Fig 2A and B should be consolidated and represented in a more comprehensive and convincing way.

Response: To address the reviewer's suggestion, we have revised Fig. 2B using violin plots followed by paired t-test statistical analysis.

Revised Fig. 2B. The oncolytic effects of M1-GFP and M1-N3E2M in different types of cancer cell lines were analyzed by a paired t test. The data are shown as violin plots.

3. The conclusion that digestive system tumors are more sensitive to the virus variant is somehow misleading. The cell lines tested are mostly of these types and not all of them are more sensitive. The virus does not seem to be specifically more sensitive for digestive system tumors. The text and conclusions should be reworded to reflect that the virus is more potent in other tumor types, not just in specific cancers.

Response: We apologize for the somehow misleading conclusion and have revised the title, abstract (line 42 of page 3), introduction (line 94 of page 5), result (line 154 of page 8) and discussion (line 338 of page 16).

4. Fig 3. What is the rationale for 21 daily injections? This seems like a lot and no justification is provided.

Response: We tested the therapeutic effect of different dosage regimens (21/6/3 daily injections) in a subcutaneous tumor model derived from HCT-116 cells in nude mice. For M1-GFP, only 21 injections led to slight but significant inhibition of tumor growth, while 6 and 3 injections had no effect. For M1-N3E2M, both 21 and 6 injections exhibited similar antitumor effect, whereas 3 injections failed to suppress tumor growth. These results indicate that the optimal dosage regimen for M1-N3E2M is 6 daily injections. The following figure shows the tumor growth after three days of continuous administration.

Figure to reviewer. HCT-116 xenografts were treated intravenously with vehicle, M1-GFP or M1-N3E2M for 3 consecutive days. Tumor volume were measured every three days. n=7; mean±SD; ns, not significant.

5. Figs 3C, H and K should be moved to supplemental. Or at least H and K since it is very repetitive.

Response: We agree with the editor and have moved Fig. 3C, H and K to the supplemental materials (Fig. S2A-C).

Fig. S2. The body weights of mice in vivo. (A) Relative body weights of mice in Fig. 3A were measured every three days. $n=7$; mean \pm SD; *ns*, not significant. (B) Relative body weights of mice in Fig. 3E were measured every three days. $n=7$; mean \pm SD; *ns*, not significant. (C) Relative body weights of mice in Fig. 3G were measured every three days. $n=10$; *ns*, not significant.

6. Figs. 3J, L and M show the same thing. L and M should be moved to supplemental.

Response: We agree with the editor and have moved Fig. 3L and M to supplemental materials (Fig. S3).

Fig. S3. Tumor weights of SW620 xenograft models. (A) Images of tumors from every group in Fig. 3H at the experimental endpoints. (B) Bar graph of tumor weights in every group at the experimental endpoints. $n=10$; mean \pm SD; *** $P<0.001$.

7. Unless I am mistaken, 3N had replicates. Why no error bars? Also, why 10% as the cutoff? This affects the interpretation and is not justified.

Response: We apologize for the confusion. We applied the Tumor histoculture end-

point staining computer image analysis (TECIA) method published previously to measure the oncolytic effect of M1 viruses in patient explants. Since the TECIA system only reports the mean value rather than the individual value of replicates for each group, the bar chart does not have error bars⁷. We selected 10% as the cutoff based on the following references⁸⁻¹².

8. Fig. 4C should be moved to supplemental.

Response: We agree with the editor and have moved Fig. 4C to supplemental materials (Fig. S4).

Fig. S4. The replication of M1 viruses in HCT-116 cells. The structural protein E1 was detected by Western blotting in HCT-116 cells infected with M1-GFP, M1-NS3M, M1-E2M, or M1-N3E2M at an MOI of 1 at 48 hpi (left). Quantification of E1 expression is shown (right). n = 3; mean±SD.

9. Fig 4D: virus output does not increase over time. Does virus production stop after 24h? Please explain this.

Response: We apologize for the confusion caused by the inappropriate presentation of virus growth kinetics using a bar chart. We have resolved this issue by replacing the original Fig. 4D (bar chart) with the revised Fig. 4C (line chart) depicting the continuous virus growth within 72h.

Revised Fig. 4C. The viral titer was determined by a TCID50 assay. We performed a statistical analysis of the final virus production. n=3; mean±SD; **P*<0.05.

10. Fig. 5D is missing a control input.

Response: We appreciate the reviewer's suggestion and have included the control input in the revised Fig 5D.

D

Revised Fig 5D. Western blot analysis of M1-GFP and M1-E2M incubated with MXRA8-His bound to His-Tag Mouse mAb Sepharose Beads. Precipitated viral particles were detected using an anti-E1 mAb (left). Quantification of E1 expression is shown (right). n=3; mean±SD; ***P*<0.01.

11. Fig. 6H, why is there a signal for PKR in all conditions (including control). And why is there an NSP3 signal in the Ig control pulldown?

Response: In the original experiment, we transiently transfected plasmids encoding nsP3 proteins into HCT-116 cells. We hypothesized that the unintentional PKR and NSP3 signal might result from the low nsP3 expression level due to low transfection efficiency. To verify this hypothesis, we generated HCT-116 cell lines stably expressing wild-type or mutant nsP3 protein by lentivirus transduction and puromycin selection. The revised Fig. 6H showed significantly higher signal-to-noise ratio.

Revised Fig. 6H. HCT-116 cell lines with stable expression of wild-type and mutant nsP3 protein were cultured in the presence of puromycin. Co-immunoprecipitation was conducted with an anti-Flag antibody or isotype control IgG prior to immunoblot analysis with anti-nsP3 and anti-PKR antibodies.

12. “Neither the expression nor the phosphorylation level of STAT1 was changed at 4 hours after infection”. Between the 2 viruses? Or with control? There is a signal for both viruses.

Response: We apologize for the confusion caused by the misleading description. We have modified the description as “Between the two viruses, neither the expression nor the phosphorylation level of STAT1 was changed at 4 hours after infection” in line 283 of page 14 in the revised manuscript.

13. Fig 7A. How was the dose selected? Based on previous studies? Please explain in the text.

Response: We appreciate the reviewer’s suggestion. We chose the dose based on the following considerations: (1) Previous study. We had intravenously administered three rounds of wild-type M1 virus (1×10^9 PFU/dose, six doses/round) in cynomolgus macaques for safety evaluation. The results support a high safety profile of M1 virus. (2) Clinical trial. Oncolytic virus M1 is undergoing a clinical trial, and the dosage used is 1×10^9 CCID50 once daily on day 1-5, every 28-day cycle. We increased the dosage and shortened the time period slightly (2×10^9 PFU once daily on day 1-5, every 19-day cycle) for safety evaluation of the mutant virus to provide flexibility for future design of clinical trials. We have added this information in the MATERIALS AND METHODS section (line 595-601 of page 27-28 in the revised manuscript).

Reviewer #4:

1. The therapeutic potential of M1-N3E2M was tested against the HCT-116 tumor in Nude mice with different frequency of the OV's administered (Fig 3A vs 3F). Obviously, the effect of the low frequency of administration of OV's (3A) gave similar results as the more frequent administration. Wouldn't it make sense to further decrease the number of OV's administration or is there a minimum number of OV's administration required to see an effect?

Response: We appreciate the reviewer's suggestion. Reviewer #3 raised a similar question and here is our response: We tested the therapeutic effect of different dosage regimens (21/6/3 daily injections) in a subcutaneous tumor model derived from HCT-116 cells in nude mice. For M1-GFP, only 21 injections led to slight but significant inhibition of tumor growth, while 6 and 3 injections had no effect. For M1-N3E2M, both 21 and 6 injections exhibited similar antitumor effect, whereas 3 injections failed to suppress tumor growth. These results indicate that the optimal dosage regimen for M1-N3E2M is 6 daily injections. The following figure shows the tumor growth after three days of continuous administration.

Figure to reviewer. HCT-116 xenografts were treated intravenously with vehicle, M1-GFP or M1-N3E2M for 3 consecutive days. Tumor volume were measured every three days. n=7; mean±SD; ns, not significant.

2. There was a significant decrease in tumor sizes after M1-N3E2M treatment in HCT-116 mice (Fig 3B and G) and in SW620 mice (3J) while the animals did not show any weight loss (3C, H and K). How do the authors explain that there was a reduction of tumor weight from 1.5 – 2 gr to approx. 0.2 gr (Fig 3M) while the weight of the animals remains around 18-20 grams (Fig 3K) and did not change over time. A reduction of 2 grams tumor weight in animals of approx. 20 grams is a reduction of 10%! Maybe the authors can give changes in body weight as a percentage rather than the actual body weights.

Response: We appreciate the reviewer’s comment. We agree that the reduction of tumor weight in M1-N3E2M-treated mice is remarkable. However, we did not observe significant changes in body weight over time in these mice. One possible explanation is that the tumor weight reduction was compensated by the increase of other tissues or organs due to the better health status. We have also changed the presentation of body weight data from absolute values to percentage changes relative to baseline (Fig 3C, H and K are moved to the revised Fig. S2).

Fig. S2. The body weights of mice in vivo. (A) Relative body weights of mice in Fig. 3A were measured every three days. n=7; mean±SD; ns, not significant. (B) Relative body weights of mice in Fig. 3E were measured every three days. n=7; mean±SD; ns, not significant. (C) Relative body weights of mice in Fig. 3G were measured every three days. n=10; ns, not significant. (D) Relative body weights of

mice in Fig. 3I were measured. $n=5$; mean \pm SD; ns, not significant. (C) Relative body weights of mice in Fig. 3K were measured every three days Control: $n=7$; M1-GFP: $n=9$; M1-N3E2M: $n=8$; ns, not significant.

3. Why is the frequency of OV_s (3 \times 5 injections) in macaques different that those in Nude mice studies? Is this administration strategy based on potential clinical trials? Please explain.

Response: We agree with the reviewer that potential clinical trials represent a crucial factor to consider when designing an administration strategy in macaques. Reviewer #3 raised a similar question and here is our response: We chose the dose based on the following considerations: (1) Previous study. We had intravenously administered three rounds of wild-type M1 virus (1×10^9 PFU/dose, six doses/round) in cynomolgus macaques for safety evaluation. The results support a high safety profile of M1 virus. (2) Clinical trial. Oncolytic virus M1 is undergoing a clinical trial, and the dosage used is 1×10^9 CCID₅₀ once daily on day 1-5, every 28-day cycle.

We increased the dosage and shortened the time period slightly (2×10^9 PFU once daily on day 1-5, every 19-day cycle) for safety evaluation of the mutant virus so that we can provide flexibility for future design of clinical trials. We have added this information in the MATERIALS AND METHODS section (line 595-601 of page 27-28 in the revised manuscript).

References:

1. Garijo R, Hernandez-Alonso P, Rivas C, Diallo JS, Sanjuan R. Experimental evolution of an oncolytic vesicular stomatitis virus with increased selectivity for p53-deficient cells. *PLoS One* 2014;9:e102365.
2. Fang L, Tian W, Zhang C, Wang X, Li W, Zhang Q, Zhang Y, Zheng J. Oncolytic adenovirus-mediated expression of CCL5 and IL12 facilitates CA9-targeting CAR-T therapy against renal cell carcinoma. *Pharmacol Res* 2023;189:106701.
3. Gahr S, Casoni GP, Falk-Paulsen M, Maschkowitz G, Bryceson YT, Voss M. Viral host range factors antagonize pathogenic SAMD9 and SAMD9L variants. *Exp Cell Res* 2023;425:113541.
4. Franco JH, Chattopadhyay S, Pan ZK. How Different Pathologies Are Affected by IFIT Expression. *Viruses* 2023;15.
5. Villamayor L, Rivero V, Lopez-Garcia D, Topham DJ, Martinez-Sobrido L, Nogales A, DeDiego ML. Interferon alpha inducible protein 6 is a negative regulator of innate immune responses by modulating RIG-I activation. *Front Immunol* 2023;14:1105309.
6. Confort MP, Duboeuf M, Thiesson A, Pons L, Marziali F, Desloire S, Ratniner M, Cimarelli A, Arnaud F. IFITMs from Naturally Infected Animal Species Exhibit Distinct Restriction Capacities against Toscana and Rift Valley Fever Viruses. *Viruses* 2023;15.
7. Lee SY, Jeon DG, Cho WH, Song WS, Kim MB, Park JH. Preliminary study of chemosensitivity tests in osteosarcoma using a histoculture drug response assay. *Anticancer Res* 2006;26:2929-32.
8. Lin Y, Zhang H, Liang J, Li K, Zhu W, Fu L, Wang F, Zheng X, Shi H, Wu S, Xiao X, Chen L, Tang L, Yan M, Yang X, Tan Y, Qiu P, Huang Y, Yin W, Su X, Hu H, Hu J, Yan G. Identification and characterization of alphavirus M1 as a selective oncolytic virus targeting ZAP-defective human cancers. *Proc Natl Acad Sci U S A* 2014;111:E4504-12.
9. Furukawa T, Kubota T, Hoffman RM. Clinical applications of the histoculture drug response assay. *Clin Cancer Res* 1995;1:305-11.
10. Cheng C, Liu ZG, Zhang H, Xie JD, Chen XG, Zhao XQ, Wang F, Liang YJ, Chen LK, Singh S, Chen JJ, Talele TT, Chen ZS, Zhong FT, Fu LW. Enhancing chemosensitivity in ABCB1- and ABCG2-overexpressing cells and cancer stem-like cells by an Aurora kinase inhibitor CCT129202. *Mol Pharm* 2012;9:1971-82.
11. Liang J, Guo L, Li K, Xiao X, Zhu W, Zheng X, Hu J, Zhang H, Cai J, Yu Y, Tan Y, Li C, Liu X, Hu C, Liu Y, Qiu P, Su X, He S, Lin Y, Yan G. Inhibition of the mevalonate pathway enhances cancer cell oncolysis mediated by M1 virus. *Nat Commun* 2018;9:1524.
12. Zhang H, Li K, Lin Y, Xing F, Xiao X, Cai J, Zhu W, Liang J, Tan Y, Fu L, Wang F, Yin W, Lu B, Qiu P, Su X, Gong S, Bai X, Hu J, Yan G. Targeting VCP enhances anticancer activity of oncolytic virus M1 in hepatocellular carcinoma. *Sci Transl Med* 2017;9.

Reviewers' Comments:

Reviewer #1:

Remarks to the Author:

A number of new experiments are included that support the original in vitro data package. The modifications to the manuscript have strengthened the study. The in vivo data is still not as compelling as many other OV systems that require only 1-3 doses to see complete responses in multiple tumour models.

Reviewer #2:

Remarks to the Author:

The authors have diligently addressed all of the reviewer comments to the extent possible.

Reviewer #4:

Remarks to the Author:

The authors have replied to my questions/suggestions in a satisfied manner.